# Policy Optimization in Adversarial MDPs: Improved Exploration via Dilated Bonuses

**Haipeng Luo**[*]    **Chen-Yu Wei**[*]    **Chung-Wei Lee**
{haipengl,chenyu.wei,leechung}@usc.edu
University of Southern California

## Abstract

Policy optimization is a widely-used method in reinforcement learning. Due to its local-search nature, however, theoretical guarantees on global optimality often rely on extra assumptions on the Markov Decision Processes (MDPs) that bypass the challenge of global exploration. To eliminate the need of such assumptions, in this work, we develop a general solution that adds *dilated bonuses* to the policy update to facilitate global exploration. To showcase the power and generality of this technique, we apply it to several episodic MDP settings with adversarial losses and bandit feedback, improving and generalizing the state-of-the-art. Specifically, in the tabular case, we obtain $\widetilde{\mathcal{O}}(\sqrt{T})$ regret where $T$ is the number of episodes, improving the $\widetilde{\mathcal{O}}(T^{2/3})$ regret bound by [27]. When the number of states is infinite, under the assumption that the state-action values are linear in some low-dimensional features, we obtain $\widetilde{\mathcal{O}}(T^{2/3})$ regret with the help of a simulator, matching the result of [24] while importantly removing the need of an exploratory policy that their algorithm requires. To our knowledge, this is the first algorithm with sublinear regret for linear function approximation with adversarial losses, bandit feedback, and no exploratory assumptions. Finally, we also discuss how to further improve the regret or remove the need of a simulator using dilated bonuses, when an exploratory policy is available.[1]

## 1   Introduction

Policy optimization methods are among the most widely-used methods in reinforcement learning. Its empirical success has been demonstrated in various domains such as computer games [26] and robotics [21]. However, due to its local-search nature, global optimality guarantees of policy optimization often rely on unrealistic assumptions to ensure global exploration (see e.g., [1, 3, 24, 30]), making it theoretically less appealing compared to other methods.

Motivated by this issue, a line of recent works [7, 27, 2, 35] equip policy optimization with global exploration by adding exploration bonuses to the update, and prove favorable guarantees even without making extra exploratory assumptions. Moreover, they all demonstrate some robustness aspect of policy optimization (such as being able to handle adversarial losses or a certain degree of model mis-specification). Despite these important progresses, however, many limitations still exist, including worse regret rates comparing to the best value-based or model-based approaches [27, 2, 35], or requiring full-information feedback on the entire loss function (as opposed to the more realistic bandit feedback) [7].

---

[*]Equal contribution.

[1]In an improved version of this paper, we show that under the linear MDP assumption, an exploratory policy is not even needed. See https://arxiv.org/abs/2107.08346.

35th Conference on Neural Information Processing Systems (NeurIPS 2021).

To address these issues, in this work, we propose a new type of exploration bonuses called *dilated bonuses*, which satisfies a certain *dilated Bellman equation* and provably leads to improved exploration compared to existing works (Section 3). We apply this general idea to advance the state-of-the-art of policy optimization for learning finite-horizon episodic MDPs with *adversarial losses and bandit feedback*. More specifically, our main results are:

- First, in the tabular setting, addressing the main open question left in [27], we improve their $\widetilde{\mathcal{O}}(T^{2/3})$ regret to the optimal $\widetilde{\mathcal{O}}(\sqrt{T})$ regret. This shows that policy optimization, which performs local optimization, is as capable as other occupancy-measure-based global optimization algorithms [15, 20] in terms of global exploration. Moreover, our algorithm is computationally more efficient than those global methods since they require solving some convex optimization in each episode. (Section 4)

- Second, to further deal with large-scale problems, we consider a linear function approximation setting where the state-action values are linear in some known low-dimensional features and also a simulator is available, the same setting considered by [24]. We obtain the same $\widetilde{\mathcal{O}}(T^{2/3})$ regret while importantly removing the need of an exploratory policy that their algorithm requires. Unlike the tabular setting (where we improve existing regret rates of policy optimization), note that researchers have not been able to show *any* sublinear regret for policy optimization without exploratory assumptions for this problem, which shows the critical role of our proposed dilated bonuses. In fact, there are simply no existing algorithms with sublinear regret *at all* for this setting, be it policy-optimization-type or not. This shows the advantage of policy optimization over other approaches, when combined with our dilated bonuses. (Section 5)

- Finally, while the main focus of our work is to show how dilated bonuses are able to provide global exploration, we also discuss their roles in improving the regret rate to $\widetilde{\mathcal{O}}(\sqrt{T})$ in the linear setting above or removing the need of a simulator for the special case of linear MDPs (with $\widetilde{\mathcal{O}}(T^{6/7})$ regret), when an exploratory policy is available. (Section 6)

**Related work.** In the tabular setting, except for [27], most algorithms apply the occupancy-measure-based framework to handle adversarial losses (e.g., [25, 15, 9, 8]), which as mentioned is computationally expensive. For stochastic losses, there are many more different approaches such as model-based ones [13, 10, 5, 12, 34] and value-based ones [14, 11].

Theoretical studies for linear function approximation have gained increasing interest recently [32, 33, 16]. Most of them study stochastic/stationary losses, with the exception of [24, 7]. Our algorithm for the linear MDP setting bears some similarity to those of [2, 35] which consider stationary losses. However, our algorithm and analysis are arguably simpler than theirs. Specifically, they divide the state space into a known part and an unknown part, with different exploration principle and bonus design for different parts. In contrast, we enjoy a unified bonus design for all states. Besides, in each episode, their algorithms first execute an exploratory policy (from a *policy cover*), and then switch to the policy suggested by the policy optimization algorithm, which inevitably leads to linear regret when facing adversarial losses.

## 2   Problem Setting

We consider an MDP specified by a state space $X$ (possibly infinite), a finite action space $A$, and a transition function $P$ with $P(\cdot|x,a)$ specifying the distribution of the next state after taking action $a$ in state $x$. In particular, we focus on the *finite-horizon episodic setting* in which $X$ admits a layer structure and can be partitioned into $X_0, X_1, \ldots, X_H$ for some fixed parameter $H$, where $X_0$ contains only the initial state $x_0$, $X_H$ contains only the terminal state $x_H$, and for any $x \in X_h$, $h = 0, \ldots, H-1$, $P(\cdot|x,a)$ is supported on $X_{h+1}$ for all $a \in A$ (that is, transition is only possible from $X_h$ to $X_{h+1}$). An episode refers to a trajectory that starts from $x_0$ and ends at $x_H$ following some series of actions and the transition dynamic. The MDP may be assigned with a loss function $\ell : X \times A \to [0,1]$ so that $\ell(x,a)$ specifies the loss suffered when selecting action $a$ in state $x$.

A policy $\pi$ for the MDP is a mapping $X \to \Delta(A)$, where $\Delta(A)$ denotes the set of distributions over $A$ and $\pi(a|x)$ is the probability of choosing action $a$ in state $x$. Given a loss function $\ell$

and a policy $\pi$, the expected total loss of $\pi$ is given by $V^\pi(x_0; \ell) = \mathbb{E}\big[\sum_{h=0}^{H-1} \ell(x_h, a_h) \mid a_h \sim \pi_t(\cdot|x_h), x_{h+1} \sim P(\cdot|x_h, a_h)\big]$. It can also be defined via the Bellman equation involving the *state value function* $V^\pi(x; \ell)$ and the *state-action value function* $Q^\pi(x, a; \ell)$ (a.k.a. $Q$-function) defined as below: $V(x_H; \ell) = 0$,

$$Q^\pi(x, a; \ell) = \ell(x, a) + \mathbb{E}_{x' \sim P(\cdot|x,a)}\left[V^\pi(x'; \ell)\right], \text{ and } V^\pi(x; \ell) = \mathbb{E}_{a \sim \pi(\cdot|x)}\left[Q^\pi(x, a; \ell)\right].$$

We study online learning in such a finite-horizon MDP with *unknown transition*, *bandit feedback*, and *adversarial losses*. The learning proceeds through $T$ episodes. Ahead of time, an adversary arbitrarily decides $T$ loss functions $\ell_1, \ldots, \ell_T$, without revealing them to the learner. Then in each episode $t$, the learner decides a policy $\pi_t$ based on all information received prior to this episode, executes $\pi_t$ starting from the initial state $x_0$, generates and observes a trajectory $\{(x_{t,h}, a_{t,h}, \ell_t(x_{t,h}, a_{t,h}))\}_{h=0}^{H-1}$. Importantly, the learner does not observe any other information about $\ell_t$ (a.k.a. bandit feedback).[2] The goal of the learner is to minimize the regret, defined as

$$\text{Reg} = \sum_{t=1}^{T} V_t^{\pi_t}(x_0) - \min_\pi \sum_{t=1}^{T} V_t^\pi(x_0),$$

where we use $V_t^\pi(x)$ as a shorthand for $V^\pi(x; \ell_t)$ (and similarly $Q_t^\pi(x, a)$ as a shorthand for $Q^\pi(x, a; \ell_t)$). Without further structures, the best existing regret bound is $\widetilde{\mathcal{O}}(H|X|\sqrt{|A|T})$ [15], with an extra $\sqrt{X}$ factor compared to the best existing lower bound [14].

**Occupancy measures.** For a policy $\pi$ and a state $x$, we define $q^\pi(x)$ to be the probability (or probability measure when $|X|$ is infinite) of visiting state $x$ within an episode when following $\pi$. When it is necessary to highlight the dependence on the transition, we write it as $q^{P,\pi}(x)$. Further define $q^\pi(x, a) = q^\pi(x)\pi(a|x)$ and $q_t(x, a) = q^{\pi_t}(x, a)$. Finally, we use $q^\star$ as a shorthand for $q^{\pi^\star}$ where $\pi^\star \in \text{argmin}_\pi \sum_{t=1}^{T} V_t^\pi(x_0)$ is one of the optimal policies.

Note that by definition, we have $V^\pi(x_0; \ell) = \sum_{x,a} q^\pi(x, a)\ell(x, a)$. In fact, we will overload the notation and let $V^\pi(x_0; b) = \sum_{x,a} q^\pi(x, a)b(x, a)$ for any function $b : X \times A \to \mathbb{R}$ (even though it might not correspond to a real loss function).

**Other notations.** We denote by $\mathbb{E}_t[\cdot]$ and $\text{Var}_t[\cdot]$ the expectation and variance conditioned on everything prior to episode $t$. For a matrix $\Sigma$ and a vector $z$ (of appropriate dimension), $\|z\|_\Sigma$ denotes the quadratic norm $\sqrt{z^\top \Sigma z}$. The notation $\widetilde{\mathcal{O}}(\cdot)$ hides all logarithmic factors.

## 3 Dilated Exploration Bonuses

In this section, we start with a general discussion on designing exploration bonuses (not specific to policy optimization), and then introduce our new dilated bonuses for policy optimization. For simplicity, the exposition in this section assumes a finite state space, but the idea generalizes to an infinite state space.

When analyzing the regret of an algorithm, very often we run into the following form:

$$\text{Reg} = \sum_{t=1}^{T} V_t^{\pi_t}(x_0) - \sum_{t=1}^{T} V_t^{\pi^\star}(x_0) \leq o(T) + \sum_{t=1}^{T} \sum_{x,a} q^\star(x, a)b_t(x, a) = o(T) + \sum_{t=1}^{T} V^{\pi^\star}(x_0; b_t),$$
$$(1)$$

for some function $b_t(x, a)$ usually related to some estimation error or variance that can be prohibitively large. For example, in policy optimization, the algorithm performs local search in each state essentially using a multi-armed bandit algorithm and treating $Q^{\pi_t}(x, a)$ as the loss of action $a$ in state $x$. Since $Q^{\pi_t}(x, a)$ is unknown, however, the algorithm has to use some estimator of $Q^{\pi_t}(x, a)$ instead, whose bias and variance both contribute to the $b_t$ function. Usually, $b_t(x, a)$ is large for a rarely-visited state-action pair $(x, a)$ and is inversely related to $q_t(x, a)$, which is exactly why most analysis relies

---

[2]Full-information feedback, on the other hand, refers to the easier setting where the entire loss function $\ell_t$ is revealed to the learner at the end of episode $t$.

on the assumption that some *distribution mismatch coefficient* related to $q^\star(x,a)/q_t(x,a)$ is bounded (see e.g., [3, 31]).

On the other hand, an important observation is that while $V^{\pi^\star}(x_0; b_t)$ can be prohibitively large, its counterpart with respect to the learner's policy $V^{\pi_t}(x_0; b_t)$ is usually nicely bounded. For example, if $b_t(x,a)$ is inversely related to $q_t(x,a)$ as mentioned, then $V^{\pi_t}(x_0; b_t) = \sum_{x,a} q_t(x,a) b_t(x,a)$ is small no matter how small $q_t(x,a)$ could be for some $(x,a)$. This observation, together with the linearity property $V^\pi(x_0; \ell_t - b_t) = V^\pi(x_0; \ell_t) - V^\pi(x_0; b_t)$, suggests that we treat $\ell_t - b_t$ as the loss function of the problem, or in other words, add a (negative) bonus to each state-action pair, which intuitively encourages exploration due to underestimation. Indeed, assuming for a moment that Eq. (1) still roughly holds even if we treat $\ell_t - b_t$ as the loss function:

$$\sum_{t=1}^{T} V^{\pi_t}(x_0; \ell_t - b_t) - \sum_{t=1}^{T} V^{\pi^\star}(x_0; \ell_t - b_t) \lesssim o(T) + \sum_{t=1}^{T} V^{\pi^\star}(x_0; b_t). \tag{2}$$

Then by linearity and rearranging, we have

$$\text{Reg} = \sum_{t=1}^{T} V_t^{\pi_t}(x_0) - \sum_{t=1}^{T} V_t^{\pi^\star}(x_0) \lesssim o(T) + \sum_{t=1}^{T} V^{\pi_t}(x_0; b_t). \tag{3}$$

Due to the switch from $\pi^\star$ to $\pi_t$ in the last term compared to Eq. (1), this is usually enough to prove a desirable regret bound without making extra assumptions.

The caveat of this discussion is the assumption of Eq. (2). Indeed, after adding the bonuses, which itself contributes some more bias and variance, one should expect that $b_t$ on the right-hand side of Eq. (2) becomes something larger, breaking the desired cancellation effect to achieve Eq. (3). Indeed, the definition of $b_t$ essentially becomes circular in this sense.

**Dilated Bonuses for Policy Optimization**  To address this issue, we take a closer look at the policy optimization algorithm specifically. As mentioned, policy optimization decomposes the problem into individual multi-armed bandit problems in each state and then performs local optimization. This is based on the well-known performance difference lemma [17]:

$$\text{Reg} = \sum_{x} q^\star(x) \sum_{t=1}^{T} \sum_{a} \Big( \pi_t(a|x) - \pi^\star(a|x) \Big) Q_t^{\pi_t}(x,a),$$

showing that in each state $x$, the learner is facing a bandit problem with $Q_t^{\pi_t}(x,a)$ being the loss for action $a$. Correspondingly, incorporating the bonuses $b_t$ for policy optimization means subtracting the bonus $Q^{\pi_t}(x,a; b_t)$ from $Q_t^{\pi_t}(x,a)$ for each action $a$ in each state $x$. Recall that $Q^{\pi_t}(x,a; b_t)$ satisfies the Bellman equation $Q^{\pi_t}(x,a; b_t) = b_t(x,a) + \mathbb{E}_{x' \sim P(\cdot|x,a)} \mathbb{E}_{a' \sim \pi_t(\cdot|x')} [B_t(x',a')]$. To resolve the issue mentioned earlier, we propose to replace this bonus function $Q^{\pi_t}(x,a; b_t)$ with its *dilated* version $B_t(s,a)$ satisfying the following *dilated Bellman equation*:

$$B_t(x,a) = b_t(x,a) + \left( 1 + \frac{1}{H} \right) \mathbb{E}_{x' \sim P(\cdot|x,a)} \mathbb{E}_{a' \sim \pi_t(\cdot|x')} [B_t(x',a')] \tag{4}$$

(with $B_t(x_H, a) = 0$ for all $a$). The only difference compared to the standard Bellman equation is the extra $\left( 1 + \frac{1}{H} \right)$ factor, which slightly increases the weight for deeper layers and thus intuitively induces more exploration for those layers. Due to the extra bonus compared to $Q^{\pi_t}(x,a; b_t)$, the regret bound also increases accordingly. In all our applications, this extra amount of regret turns out to be of the form $\frac{1}{H} \sum_{t=1}^{T} \sum_{x,a} q^\star(x) \pi_t(a|x) B_t(x,a)$, leading to

$$\sum_{x} q^\star(x) \sum_{t=1}^{T} \sum_{a} \Big( \pi_t(a|x) - \pi^\star(a|x) \Big) \Big( Q_t^{\pi_t}(x,a) - B_t(x,a) \Big)$$

$$\leq o(T) + \sum_{t=1}^{T} V^{\pi^\star}(x_0; b_t) + \frac{1}{H} \sum_{t=1}^{T} \sum_{x,a} q^\star(x) \pi_t(a|x) B_t(x,a). \tag{5}$$

With some direct calculation, one can show that this is enough to show a regret bound that is only a constant factor larger than the desired bound in Eq. (3)! This is summarized in the following lemma.

**Lemma 3.1.** *If Eq. (5) holds with $B_t$ defined in Eq. (4), then* $\text{Reg} \leq o(T) + 3 \sum_{t=1}^{T} V^{\pi_t}(x_0; b_t)$.

The high-level idea of the proof is to show that the bonuses added to a layer $h$ is enough to cancel the large bias/variance term (including those coming from the bonus itself) from layer $h + 1$. Therefore, cancellation happens in a layer-by-layer manner except for layer 0, where the total amount of bonus can be shown to be at most $(1 + \frac{1}{H})^H \sum_{t=1}^{T} V^{\pi_t}(x_0; b_t) \leq 3 \sum_{t=1}^{T} V^{\pi_t}(x_0; b_t)$.

Recalling again that $V^{\pi_t}(x_0; b_t)$ is usually nicely bounded, we thus arrive at a favorable regret guarantee without making extra assumptions. Of course, since the transition is unknown, we cannot compute $B_t$ exactly. However, Lemma 3.1 is robust enough to handle either a good approximate version of $B_t$ (see Lemma B.1) or a version where Eq. (4) and Eq. (5) only hold in expectation (see Lemma B.2), which is enough for us to handle unknown transition. In the next three sections, we apply this general idea to different settings, showing what $b_t$ and $B_t$ are concretely in each case.

## 4 The Tabular Case

In this section, we study the tabular case where the number of states is finite. We propose a policy optimization algorithm with $\widetilde{\mathcal{O}}(\sqrt{T})$ regret, improving the $\widetilde{\mathcal{O}}(T^{2/3})$ regret of [27]. See Algorithm 1 for the complete pseudocode.

**Algorithm design.** First, to handle unknown transition, we follow the common practice (dating back to [13]) to maintain a confidence set of the transition, which is updated whenever the visitation count of a certain state-action pair is doubled. We call the period between two model updates an epoch, and use $\mathcal{P}_k$ to denote the confidence set for epoch $k$, formally defined in Eq. (10).

In episode $t$, the policy $\pi_t$ is defined via the standard multiplicative weight algorithm (also connected to Natural Policy Gradient [18, 3, 30]), but importantly with the dilated bonuses incorporated such that $\pi_t(a|x) \propto \exp(-\eta \sum_{\tau=1}^{t-1}(\widehat{Q}_\tau(x, a) - B_\tau(x, a)))$. Here, $\eta$ is a step size parameter, $\widehat{Q}_\tau(x, a)$ is an importance-weighted estimator for $Q_\tau^{\pi_\tau}(x, a)$ defined in Eq. (7), and $B_\tau(x, a)$ is the dilated bonus defined in Eq. (9).

More specifically, for a state $x$ in layer $h$, $\widehat{Q}_t(x, a)$ is defined as $\frac{L_{t,h} \mathbb{1}_t(x,a)}{\overline{q}_t(x,a)+\gamma}$, where $\mathbb{1}_t(x, a)$ is the indicator of whether $(x, a)$ is visited during episode $t$; $L_{t,h}$ is the total loss suffered by the learner starting from layer $h$ till the end of the episode; $\overline{q}_t(x, a) = \max_{\widehat{P} \in \mathcal{P}_k} q^{\widehat{P}, \pi_t}(x, a)$ is the largest plausible value of $q_t(x, a)$ within the confidence set, which can be computed efficiently using the COMP-UOB procedure of [15] (see also Appendix C.1); and finally $\gamma$ is a parameter used to control the maximum magnitude of $\widehat{Q}_t(x, a)$, inspired by the work of [23]. To get a sense of this estimator, consider the special case when $\gamma = 0$ and the transition is known so that we can set $\mathcal{P}_k = \{P\}$ and thus $\overline{q}_t = q_t$. Then, since the expectation of $L_{t,h}$ conditioned on $(x, a)$ being visited is $Q_t^{\pi_t}(x, a)$ and the expectation of $\mathbb{1}_t(x, a)$ is $q_t(x, a)$, we know that $\widehat{Q}_t(x, a)$ is an unbiased estimator for $Q_t^{\pi_t}(x, a)$. The extra complication is simply due to the transition being unknown, forcing us to use $\overline{q}_t$ and $\gamma > 0$ to make sure that $\widehat{Q}_t(x, a)$ is an optimistic underestimator, an idea similar to [15].

Next, we explain the design of the dilated bonus $B_t$. Following the discussions of Section 3, we first figure out what the corresponding $b_t$ function is in Eq. (1), by analyzing the regret bound without using any bonuses. The concrete form of $b_t$ turns out to be Eq. (8), whose value at $(x, a)$ is independent of $a$ and thus written as $b_t(x)$ for simplicity. Note that Eq. (8) depends on the occupancy measure lower bound $\underline{q}_t(s, a) = \min_{\widehat{P} \in \mathcal{P}_k} q^{\widehat{P}, \pi_t}(x, a)$, the opposite of $\overline{q}_t(s, a)$, which can also be computed efficiently using a procedure similar to COMP-UOB (see Appendix C.1). Once again, to get a sense of this, consider the special case with a known transition so that we can set $\mathcal{P}_k = \{P\}$ and thus $\overline{q}_t = \underline{q}_t = q_t$. Then, one see that $b_t(x)$ is simply upper bounded by $\mathbb{E}_{a \sim \pi_t(\cdot|x)}\left[\frac{3\gamma H}{q_t(x,a)}\right] = \frac{3\gamma H |A|}{q_t(x)}$, which is inversely related to the probability of visiting state $x$, matching the intuition we provided in Section 3 (that $b_t(x)$ is large if $x$ is rarely visited). The extra complication of Eq. (8) is again just due to the unknown transition.

With $b_t(x)$ ready, the final form of the dilated bonus $B_t$ is defined following the dilated Bellman equation of Eq. (4), except that since $P$ is unknown, we once again apply optimism and find the

---

[3] We use $y \xleftarrow{+} z$ as a shorthand for the increment operation $y \leftarrow y + z$.

---
**Algorithm 1** Policy Optimization with Dilated Bonuses (Tabular Case)
---
**Parameters:** $\delta \in (0,1)$, $\eta = \min\left\{\frac{1}{24H^3}, \frac{1}{\sqrt{|X||A|HT}}\right\}$, $\gamma = 2\eta H$.

**Initialization:** Set epoch index $k = 1$ and confidence set $\mathcal{P}_1$ as the set of all transition functions. For all $(x, a, x')$, initialize counters $N_0(x, a) = N_1(x, a) = 0$, $N_0(x, a, x') = N_1(x, a, x') = 0$.

**for** $t = 1, 2, \ldots, T$ **do**

> **Step 1: Compute and execute policy.** Execute $\pi_t$ for one episode, where
>
> $$\pi_t(a|x) \propto \exp\left(-\eta \sum_{\tau=1}^{t-1}\left(\widehat{Q}_\tau(x, a) - B_\tau(x, a)\right)\right), \tag{6}$$
>
> and obtain trajectory $\{(x_{t,h}, a_{t,h}, \ell_t(x_{t,h}, a_{t,h}))\}_{h=0}^{H-1}$.
>
> **Step 2: Construct $Q$-function estimators.** For all $h \in \{0, \ldots, H-1\}$ and $(x, a) \in X_h \times A$,
>
> $$\widehat{Q}_t(x, a) = \frac{L_{t,h}}{\overline{q}_t(x, a) + \gamma} \mathbb{1}_t(x, a), \tag{7}$$
>
> with $L_{t,h} = \sum_{i=h}^{H-1} \ell_t(x_{t,i}, a_{t,i}), \overline{q}_t(x, a) = \max_{\widehat{P} \in \mathcal{P}_k} q^{\widehat{P}, \pi_t}(x, a), \mathbb{1}_t(x, a) = \mathbb{1}\{x_{t,h} = x, a_{t,h} = a\}$.
>
> **Step 3: Construct bonus functions.** For all $(x, a) \in X \times A$,
>
> $$b_t(x) = \mathbb{E}_{a \sim \pi_t(\cdot|x)}\left[\frac{3\gamma H + H(\overline{q}_t(x, a) - \underline{q}_t(x, a))}{\overline{q}_t(x, a) + \gamma}\right] \tag{8}$$
>
> $$B_t(x, a) = b_t(x) + \left(1 + \frac{1}{H}\right) \max_{\widehat{P} \in \mathcal{P}_k} \mathbb{E}_{x' \sim \widehat{P}(\cdot|x,a)} \mathbb{E}_{a' \sim \pi_t(\cdot|x')}[B_t(x', a')] \tag{9}$$
>
> where $\underline{q}_t(x, a) = \min_{\widehat{P} \in \mathcal{P}_k} q^{\widehat{P}, \pi_t}(x, a)$ and $B_t(x_H, a) = 0$ for all $a$.
>
> **Step 4: Update model estimation.** $\forall h < H$, $N_k(x_{t,h}, a_{t,h}) \overset{+}{\leftarrow} 1$, $N_k(x_{t,h}, a_{t,h}, x_{t,h+1}) \overset{+}{\leftarrow} 1$.[3]
>
> **if** $\exists h$, $N_k(x_{t,h}, a_{t,h}) \geq \max\{1, 2N_{k-1}(x_{t,h}, a_{t,h})\}$ **then**
>
> > Increment epoch index $k \overset{+}{\leftarrow} 1$ and copy counters: $N_k \leftarrow N_{k-1}$, $N_k \leftarrow N_{k-1}$.
> >
> > Compute empirical transition $\overline{P}_k(x'|x, a) = \frac{N_k(x, a, x')}{\max\{1, N_k(x, a)\}}$ and confidence set:
> >
> > $$\begin{aligned} \mathcal{P}_k = \Big\{ &\widehat{P} : \left|\widehat{P}(x'|x, a) - \overline{P}_k(x'|x, a)\right| \leq conf_k(x'|x, a), \\ &\forall (x, a, x') \in X_h \times A \times X_{h+1}, h = 0, 1, \ldots, H-1 \Big\}, \end{aligned} \tag{10}$$
> >
> > where $conf_k(x'|x, a) = 4\sqrt{\frac{\overline{P}_k(x'|x,a) \ln\left(\frac{T|X||A|}{\delta}\right)}{\max\{1, N_k(x, a)\}}} + \frac{28 \ln\left(\frac{T|X||A|}{\delta}\right)}{3 \max\{1, N_k(x, a)\}}$.

---

largest possible value within the confidence set (see Eq. (9)). This can again be efficiently computed; see Appendix C.1. This concludes the complete algorithm design.

**Regret analysis.** The regret guarantee of Algorithm 1 is presented below:

**Theorem 4.1.** *Algorithm 1 ensures that with probability* $1 - \mathcal{O}(\delta)$, $\text{Reg} = \widetilde{\mathcal{O}}\left(H^2|X|\sqrt{AT} + H^4\right)$.

Again, this improves the $\widetilde{\mathcal{O}}(T^{2/3})$ regret of [27]. It almost matches the best existing upper bound for this problem, which is $\widetilde{\mathcal{O}}(H|X|\sqrt{|A|T})$ [15]. While it is unclear to us whether this small gap can be closed using policy optimization, we point out that our algorithm is arguably more efficient than that of [15], which performs global convex optimization over the set of all plausible occupancy measures in each episode.

The complete proof of this theorem is deferred to Appendix C. Here, we only sketch an outline of proving Eq. (5), which, according to the discussions in Section 3, is the most important part of the analysis. Specifically, we decompose the left-hand side of Eq. (5), $\sum_x q^\star(x) \sum_t \langle \pi_t(\cdot|x) - \pi^\star(\cdot|x), Q_t(x, \cdot) - B_t(x, \cdot) \rangle$, as BIAS-1 + BIAS-2 + REG-TERM, where

- BIAS-1 $= \sum_x q^\star(x) \sum_t \langle \pi_t(\cdot|x), Q_t(x, \cdot) - \widehat{Q}_t(x, \cdot) \rangle$ measures the amount of underestimation of $\widehat{Q}_t$ related to $\pi_t$, which can be bounded by $\sum_t \sum_{x,a} q^\star(x) \pi_t(a|x) \left( \frac{2\gamma H + H(\overline{q}_t(x,a) - q_t(x,a))}{\overline{q}_t(x,a) + \gamma} \right) + \widetilde{\mathcal{O}}\left( H/\eta \right)$ with high probability (Lemma C.1);

- BIAS-2 $= \sum_x q^\star(x) \sum_t \langle \pi^\star(\cdot|x), \widehat{Q}_t(x, \cdot) - Q_t(x, \cdot) \rangle$ measures the amount of overestimation of $\widehat{Q}_t$ related to $\pi^\star$, which can be bounded by $\widetilde{\mathcal{O}}\left( H/\eta \right)$ since $\widehat{Q}_t$ is an underestimator (Lemma C.2);

- REG-TERM $= \sum_x q^\star(x) \sum_t \langle \pi_t(\cdot|x) - \pi^\star(\cdot|x), \widehat{Q}_t(x, \cdot) - B_t(x, \cdot) \rangle$ is directly controlled by the multiplicative weight update, and is bounded by $\sum_t \sum_{x,a} q^\star(x) \pi_t(a|x) \left( \frac{\gamma H}{\overline{q}_t(x,a) + \gamma} + \frac{B_t(x,a)}{H} \right) + \widetilde{\mathcal{O}}\left( H/\eta \right)$ with high probability (Lemma C.3).

Combining all with the definition of $b_t$ proves the key Eq. (5) (with the $o(T)$ term being $\widetilde{\mathcal{O}}(H/\eta)$).

## 5 The Linear-$Q$ Case

In this section, we move on to the more challenging setting where the number of states might be infinite, and function approximation is used to generalize the learner's experience to unseen states. We consider the most basic linear function approximation scheme where for any $\pi$, the $Q$-function $Q_t^\pi(x, a)$ is linear in some known feature vector $\phi(x, a)$, formally stated below.

**Assumption 1** (Linear-$Q$). *Let $\phi(x, a) \in \mathbb{R}^d$ be a known feature vector of the state-action pair $(x, a)$. We assume that for any episode $t$, policy $\pi$, and layer $h$, there exists an unknown weight vector $\theta_{t,h}^\pi \in \mathbb{R}^d$ such that for all $(x, a) \in X_h \times A$, $Q_t^\pi(x, a) = \phi(x, a)^\top \theta_{t,h}^\pi$. Without loss of generality, we assume $\|\phi(x, a)\| \leq 1$ for all $(x, a)$ and $\|\theta_{t,h}^\pi\| \leq \sqrt{d}H$ for all $t, h, \pi$.*

For justification on the last condition on norms, see [30, Lemma 8]. This linear-$Q$ assumption has been made in several recent works with stationary losses [1, 30] and also in [24] with the same adversarial losses.[4] It is weaker than the linear MDP assumption (see Section 6) as it does not pose explicit structure requirements on the loss and transition functions. Due to this generality, however, our algorithm also requires access to a *simulator* to obtain samples drawn from the transition, formally stated below.

**Assumption 2** (Simulator). *The learner has access to a simulator, which takes a state-action pair $(x, a) \in X \times A$ as input, and generates a random outcome of the next state $x' \sim P(\cdot|x, a)$.*

Note that this assumption is also made by [24] and more earlier works with stationary losses (see e.g., [4, 28]).[5] In this setting, we propose a new policy optimization algorithm with $\widetilde{\mathcal{O}}(T^{2/3})$ regret. See Algorithm 2 for the pseudocode.

**Algorithm design.** The algorithm still follows the multiplicative weight update Eq. (11) in each state $x \in X_h$ (for some $h$), but now with $\phi(x, a)^\top \widehat{\theta}_{t,h}$ as an estimator for $Q_t^{\pi_t}(x, a) = \phi(x, a)^\top \theta_{t,h}^{\pi_t}$, and BONUS$(t, x, a)$ as the dilated bonus $B_t(x, a)$. Specifically, the construction of the weight estimator $\widehat{\theta}_{t,h}$ follows the idea of [24] (which itself is based on the linear bandit literature) and is defined in Eq. (12) as $\widehat{\Sigma}_{t,h}^+ \phi(x_{t,h}, a_{t,h}) L_{t,h}$. Here, $\widehat{\Sigma}_{t,h}^+$ is an $\epsilon$-accurate estimator of $(\gamma I + \Sigma_{t,h})^{-1}$, where $\gamma$ is a small parameter and $\Sigma_{t,h} = \mathbb{E}_t[\phi(x_{t,h}, a_{t,h}) \phi(x_{t,h}, a_{t,h})^\top]$ is the covariance matrix for layer $h$ under policy $\pi_t$; $L_{t,h} = \sum_{i=h}^{H-1} \ell_t(x_{t,i}, a_{t,i})$ is again the loss suffered by the learner starting from layer $h$, whose conditional expectation is $Q_t^{\pi_t}(x_{t,h}, a_{t,h}) = \phi(x_{t,h}, a_{t,h})^\top \theta_{t,h}^{\pi_t}$. Therefore,

---

[4]The assumption in [24] is stated slightly differently (e.g., their feature vectors are independent of the action). However, it is straightforward to verify that the two versions are equivalent.

[5]The simulator required by [24] is in fact slightly weaker than ours and those from earlier works — it only needs to be able to generate a trajectory starting from $x_0$ for any policy.

---

**Algorithm 2** Policy Optimization with Dilated Bonuses (Linear-$Q$ Case)

---

**parameters**: $\gamma, \beta, \eta, \epsilon$, $M = \left\lceil \frac{24 \ln(dHT)}{\epsilon^2 \gamma^2} \right\rceil$, $N = \left\lceil \frac{2}{\gamma} \ln \frac{1}{\epsilon \gamma} \right\rceil$.

**for** $t = 1, 2, \ldots, T$ **do**

    **Step 1: Interact with the environment.** Execute $\pi_t$, which is defined such that for each $x \in X_h$,

$$\pi_t(a|x) \propto \exp\left( -\eta \sum_{\tau=1}^{t-1} \left( \phi(x,a)^\top \widehat{\theta}_{\tau,h} - \textsc{Bonus}(\tau, x, a) \right) \right), \tag{11}$$

    and obtain trajectory $\{(x_{t,h}, a_{t,h}, \ell_t(x_{t,h}, a_{t,h}))\}_{h=0}^{H-1}$.

    **Step 2: Construct covariance matrix inverse estimators.**

$$\left\{ \widehat{\Sigma}_{t,h}^+ \right\}_{h=0}^{H-1} = \textsc{GeometricResampling}\,(t, M, N, \gamma)\,. \qquad \text{(see Algorithm 7)}$$

    **Step 3: Construct $Q$-function weight estimators.** For $h = 0, \ldots, H-1$, compute

$$\widehat{\theta}_{t,h} = \widehat{\Sigma}_{t,h}^+ \phi(x_{t,h}, a_{t,h}) L_{t,h}, \qquad \text{where } L_{t,h} = \sum_{i=h}^{H-1} \ell_t(x_{t,i}, a_{t,i}). \tag{12}$$

---

---

**Algorithm 3** $\textsc{Bonus}(t, x, a)$

---

**if** $\textsc{Bonus}(t, x, a)$ *has been called before* **then**

    |   **return** the value of $\textsc{Bonus}(t, x, a)$ calculated last time.

Let $h$ be such that $x \in X_h$. **if** $h = H$ **then return** $0$.

Compute $\pi_t(\cdot|x)$, defined in Eq. (11) (which involves recursive calls to $\textsc{Bonus}$ for smaller $t$).

Get a sample of the next state $x' \leftarrow \textsc{Simulator}(x, a)$.

Compute $\pi_t(\cdot|x')$ (again, defined in Eq. (11)), and sample an action $a' \sim \pi_t(\cdot|x')$.

**return** $\beta\|\phi(x,a)\|_{\widehat{\Sigma}_{t,h}^+}^2 + \mathbb{E}_{j \sim \pi_t(\cdot|x)}\left[ \beta\|\phi(x,j)\|_{\widehat{\Sigma}_{t,h}^+}^2 \right] + \left(1 + \frac{1}{H}\right) \textsc{Bonus}(t, x', a')$.

---

when $\gamma$ and $\epsilon$ approach $0$, one see that $\widehat{\theta}_{t,h}$ is indeed an unbiased estimator of $\theta_{t,h}^{\pi_t}$. We adopt the $\textsc{GeometricResampling}$ procedure (see Algorithm 7) of [24] to compute $\widehat{\Sigma}_{t,h}^+$, which involves calling the simulator multiple times.

Next, we explain the design of the dilated bonus. Again, following the general principle discussed in Section 3, we identify $b_t(x,a)$ in this case as $\beta\|\phi(x,a)\|_{\widehat{\Sigma}_{t,h}^+}^2 + \mathbb{E}_{j \sim \pi_t(\cdot|x)}\left[ \beta\|\phi(x,j)\|_{\widehat{\Sigma}_{t,h}^+}^2 \right]$ for some parameter $\beta > 0$. Further following the dilated Bellman equation Eq. (4), we thus define $\textsc{Bonus}(t, x, a)$ recursively as the last line of Algorithm 3, where we replace the expectation $\mathbb{E}_{(x',a')}[\textsc{Bonus}(t, x', a')]$ with one single sample for efficient implementation.

However, even more care is needed to actually implement the algorithm. First, since the state space is potentially infinite, one cannot actually calculate and store the value of $\textsc{Bonus}(t, x, a)$ for all $(x, a)$, but can only calculate them on-the-fly when needed. Moreover, unlike the estimators for $Q_t^{\pi_t}(x, a)$, which can be succinctly represented and stored via the weight estimator $\widehat{\theta}_{t,h}$, this is not possible for $\textsc{Bonus}(t, x, a)$ due to the lack of any structure. Even worse, the definition of $\textsc{Bonus}(t, x, a)$ itself depends on $\pi_t(\cdot|x)$ and also $\pi_t(\cdot|x')$ for the afterstate $x'$, which, according to Eq. (11), further depends on $\textsc{Bonus}(\tau, x, a)$ for $\tau < t$, resulting in a complicated recursive structure. This is also why we present it as a procedure in Algorithm 3 (instead of $B_t(x, a)$). In total, this leads to $(TAH)^{\mathcal{O}(H)}$ number of calls to the simulator. Whether this can be improved is left as a future direction.

**Regret guarantee**    By showing that Eq. (5) holds in expectation for our algorithm, we obtain the following regret guarantee. (See Appendix D for the proof.)

**Theorem 5.1.** *Under Assumption 1 and Assumption 2, with appropriate choices of the parameters $\gamma, \beta, \eta, \epsilon$, Algorithm 2 ensures $\mathbb{E}[\text{Reg}] = \widetilde{\mathcal{O}}\left(H^2(dT)^{2/3}\right)$ (the dependence on $|A|$ is only logarithmic).*

This matches the $\widetilde{\mathcal{O}}(T^{2/3})$ regret of [24, Theorem 1], without the need of their assumption which essentially says that the learner is given an exploratory policy to start with.[6] To our knowledge, this is the first no-regret algorithm for linear function approximation (with adversarial losses and bandit feedback) when no exploratory assumptions are made.

## 6 Improvements with an Exploratory Policy

Previous sections have demonstrated the role of dilated bonuses in providing global exploration. In this section, we further discuss what dilated bonuses can achieve when an exploratory policy $\pi_0$ is given in linear function approximation settings. Formally, let $\Sigma_h = \mathbb{E}[\phi(x_h, a_h)\phi(x_h, a_h)^\top]$ denote the covariance matrix for features in layer $h$ following $\pi_0$ (that is, the expectation is taken over a trajectory $\{(x_h, a_h)\}_{h=0}^{H-1}$ with $a_h \sim \pi_0(\cdot|x_h)$), then we assume the following.

**Assumption 3** (An exploratory policy). *An exploratory policy $\pi_0$ is given to the learner ahead of time, and guarantees that for any $h$, the eigenvalues of $\Sigma_h$ are at least $\lambda_{\min} > 0$.*

The same assumption is made by [24] (where they simply let $\pi_0$ be the uniform exploration policy). As mentioned, under this assumption they achieve $\widetilde{\mathcal{O}}(T^{2/3})$ regret. By slightly modifying our Algorithm 2 (specifically, executing $\pi_0$ with a small probability in each episode and setting the parameters differently), we achieve the following improved result.

**Theorem 6.1.** *Under Assumptions 1, 2, and 3, Algorithm 8 ensures* $\mathbb{E}[\text{Reg}] = \widetilde{\mathcal{O}}\big(\sqrt{\frac{H^4 T}{\lambda_{\min}}} + \sqrt{H^5 dT}\big)$.

**Removing the simulator** One drawback of our algorithm is that it requires exponential in $H$ number of calls to the simulator. To address this issue, and in fact, to also completely remove the need of a simulator, we further consider a special case where the transition function also has a low-rank structure, known as the linear MDP setting.

**Assumption 4** (Linear MDP). *The MDP satisfies Assumption 1 and that for any $h$ and $x' \in X_{h+1}$, there exists a weight vector $\nu_h^{x'} \in \mathbb{R}^d$ such that $P(x'|x, a) = \phi(x, a)^\top \nu_h^{x'}$ for all $(x, a) \in X_h \times A$.*

There is a surge of works studying this setting, with [7] being the closest to us. They achieve $\widetilde{\mathcal{O}}(\sqrt{T})$ regret but require full-information feedback of the loss functions, and there are no existing results for the bandit feedback setting without a simulator. We propose the first algorithm with sublinear regret for this problem, shown in Algorithm 10 of Appendix F due to space limit.

The structure of Algorithm 10 is very similar to that of Algorithm 2, with the same definition of $b_t(x, a)$. However, due to the low-rank transition structure, we are now able to efficiently construct estimators of $B_t(x, a)$ even for unseen state-action pairs using function approximation, bypassing the requirement of a simulator. Specifically, observe that according to Eq. (4), for each $x \in X_h$, under Assumption 4 $B_t(x, a)$ can be written as $b_t(x, a) + \phi(x, a)^\top \Lambda_{t,h}^{\pi_t}$, where $\Lambda_{t,h}^{\pi_t} = (1 + \frac{1}{H}) \int_{x' \in X_{h+1}} \mathbb{E}_{a' \sim \pi_t(\cdot|x')}[B_t(x', a')]\nu_h^{x'} dx'$ is a vector independent of $(x, a)$. Thus, by the same idea of estimating $\theta_{t,h}^{\pi_t}$, we can estimate $\Lambda_{t,h}^{\pi_t}$ as well, thus succinctly representing $B_t(x, a)$ for all $(x, a)$.

Recall that estimating $\theta_{t,h}^{\pi_t}$ (and thus also $\Lambda_{t,h}^{\pi_t}$) requires constructing the covariance matrix inverse estimate $\widehat{\Sigma}_{t,h}^+$. Due to the lack of a simulator, another important change in the algorithm is to construct $\widehat{\Sigma}_{t,h}^+$ using *online* samples. To do so, we divide the entire horizon into epochs with equal length, and only update the policy optimization algorithm at the beginning of an epoch. Within an epoch, we keep executing the same policy and collect several trajectories, which are then used to construct $\widehat{\Sigma}_{t,h}^+$. With these changes, we successfully remove the need of a simulator, and prove the guarantee below.

**Theorem 6.2.** *Under Assumption 3 and Assumption 4, Algorithm 10 ensures* $\mathbb{E}[\text{Reg}] = \widetilde{\mathcal{O}}\big(T^{6/7}\big)$ *(see Appendix F for dependence on other parameters).*

One potential direction to further improve our algorithm is to reuse data across different epochs, an idea adopted by several recent works [35, 19] for different problems. We also conjecture that

---

[6]Under an even strong assumption that every policy is exploratory, they also improve the regret to $\widetilde{\mathcal{O}}(\sqrt{T})$; see [24, Theorem 2].

Assumption 3 can be removed, but we meet some technical difficulty in proving so. We leave these for future investigation.

## Acknowledgments and Disclosure of Funding

We thank Gergely Neu and Julia Olkhovskaya for discussions on the technical details of their GEOMETRICRESAMPLING procedure. This work is supported by NSF Award IIS-1943607 and a Google Faculty Research Award.

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
