## A    Auxiliary Lemmas

In this section, we list auxiliary lemmas that are useful in our analysis. First, we show some concentration inequalities.

**Lemma A.1** ((A special form of) Freedman's inequality, Theorem 1 of [6])**.** *Let $\mathcal{F}_0 \subset \cdots \subset \mathcal{F}_n$ be a filtration, and $X_1, \ldots, X_n$ be real random variables such that $X_i$ is $\mathcal{F}_i$-measurable, $\mathbb{E}[X_i|\mathcal{F}_i] = 0$, $|X_i| \leq b$, and $\sum_{i=1}^{n} \mathbb{E}[X_i^2|\mathcal{F}_i] \leq V$ for some fixed $b \geq 0$ and $V \geq 0$. Then for any $\delta \in (0, 1)$, we have with probability at least $1 - \delta$,*

$$\sum_{i=1}^{n} X_i \leq \frac{V}{b} + b\log(1/\delta).$$

Throughout the appendix, we let $\mathcal{F}_t$ be the $\sigma$-algebra generated by the observations before episode $t$.

**Lemma A.2** (Adapted from Lemma 11 of [15]; see also [23])**.** *For all $x, a$, let $\{z_t(x, a)\}_{t=1}^{T}$ be a sequence of functions where $z_t(x, a) \in [0, R]$ is $\mathcal{F}_t$-measurable. Let $Z_t(x, a) \in [0, R]$ be a random variable such that $\mathbb{E}_t[Z_t(x, a)] = z_t(x, a)$. Then with probability at least $1 - \delta$,*

$$\sum_{t=1}^{T} \sum_{x,a} \left( \frac{\mathbb{1}_t(x,a)Z_t(x,a)}{\overline{q}_t(x,a) + \gamma} - \frac{q_t(x,a)z_t(x,a)}{\overline{q}_t(x,a)} \right) \leq \frac{RH}{2\gamma}\ln\frac{H}{\delta}.$$

**Lemma A.3** (Matrix Azuma, Theorem 7.1 of [29])**.** *Consider an adapted sequence $\{X_k\}_{k=1}^{n}$ of self-adjoint matrices in dimension $d$, and a fixed sequence $\{A_k\}_{k=1}^{n}$ of self-adjoint matrices that satisfy*

$$\mathbb{E}_k X_k = 0 \quad and \quad X_k^2 \preceq A_k^2 \text{ almost surely}$$

*Define the variance parameter*

$$\sigma^2 = \left\| \frac{1}{n} \sum_{k=1}^{n} A_k^2 \right\|_{op}.$$

*Then, for all $\tau > 0$,*

$$\Pr\left\{ \left\| \frac{1}{n} \sum_{k=1}^{n} X_k \right\|_{op} \geq \tau \right\} \leq d e^{-n\tau^2/8\sigma^2}.$$

Next, we show a classic regret bound for the exponential weight algorithm, which can be found, for example, in [22].

**Lemma A.4** (Regret bound of exponential weight, extracted from Theorem 1 of [22])**.** *Let $\eta > 0$, and let $\pi_t \in \Delta(A)$ and $\ell_t \in \mathbb{R}^A$ satisfy the following for all $t \in [T]$ and $a \in A$:*

$$\pi_1(a) = \frac{1}{|A|},$$

$$\pi_{t+1}(a) = \frac{\pi_t(a)e^{-\eta\ell_t(a)}}{\sum_{a' \in A} \pi_t(a')e^{-\eta\ell_t(a')}},$$

$$|\eta \ell_t(a)| \le 1.$$

*Then for any $\pi^\star \in \Delta(A)$,*

$$\sum_{t=1}^{T} \sum_{a \in A} (\pi_t(a) - \pi^\star(a))\ell_t(a) \le \frac{\ln |A|}{\eta} + \eta \sum_{t=1}^{T} \sum_{a \in A} \pi_t(a)\ell_t(a)^2.$$

# B    Proofs Omitted in Section 3

In this section, we prove Lemma 3.1. In fact, we prove two generalized versions of it. Lemma B.1 states that the lemma holds even when we replace the definition of $B_t(x, a)$ by an upper bound of the right hand side of Eq. (4). (Note that Lemma 3.1 is clearly a special case with $\widehat{P} = P$.)

**Lemma B.1.** *Let $b_t(x, a)$ be a non-negative loss function, and $\widehat{P}$ be a transition function. Suppose that the following holds for all $x, a$:*

$$B_t(x, a) = b_t(x, a) + \left(1 + \frac{1}{H}\right) \mathbb{E}_{x' \sim \widehat{P}(\cdot|x,a)} \mathbb{E}_{a' \sim \pi_t(\cdot|x')} \left[B_t(x', a')\right] \tag{13}$$

$$\ge b_t(x, a) + \left(1 + \frac{1}{H}\right) \mathbb{E}_{x' \sim P(\cdot|x,a)} \mathbb{E}_{a' \sim \pi_t(\cdot|x')} \left[B_t(x', a')\right]$$

*with $B_t(x_H, a) \triangleq 0$, and suppose that Eq. (5) holds. Then*

$$\mathrm{Reg} \le o(T) + 3 \sum_{t=1}^{T} \widehat{V}^{\pi_t}(x_0; b_t).$$

*where $\widehat{V}^\pi$ is the state value function under the transition function $\widehat{P}$ and policy $\pi$.*

*Proof of Lemma B.1.* By rearranging Eq. (5), we see that

$$\mathrm{Reg} \le o(T) + \underbrace{\sum_{t=1}^{T} \sum_{x,a} q^\star(x)\pi^\star(a|x)b_t(x, a)}_{\mathrm{TERM}_1}$$

$$+ \underbrace{\frac{1}{H} \sum_{t=1}^{T} \sum_{x,a} q^\star(x)\pi_t(a|x)B_t(x, a)}_{\mathrm{TERM}_2} + \underbrace{\sum_{t=1}^{T} \sum_{x,a} q^\star(x)\Big(\pi_t(a|x) - \pi^\star(a|x)\Big)B_t(x, a)}_{\mathrm{TERM}_3}.$$

We first focus on $\mathrm{TERM}_3$, and focus on a single layer $0 \le h \le H - 1$ and a single $t$:

$$\sum_{x \in X_h} \sum_{a \in A} q^\star(x) \left(\pi_t(a|x) - \pi^\star(a|x)\right) B_t(x, a)$$

$$= \sum_{x \in X_h} \sum_{a \in A} q^\star(x)\pi_t(a|x)B_t(x, a) - \sum_{x \in X_h} \sum_{a \in A} q^\star(x)\pi^\star(a|x)B_t(x, a)$$

$$= \sum_{x \in X_h} \sum_{a \in A} q^\star(x)\pi_t(a|x)B_t(x, a)$$

$$- \sum_{x \in X_h} \sum_{a \in A} q^\star(x)\pi^\star(a|x) \left(b_t(x, a) + \left(1 + \frac{1}{H}\right) \mathbb{E}_{x' \sim \widehat{P}(\cdot|x,a)} \mathbb{E}_{a' \sim \pi_t(\cdot|x')} \left[B_t(x', a')\right]\right)$$

$$\le \sum_{x \in X_h} \sum_{a \in A} q^\star(x)\pi_t(a|x)B_t(x, a)$$

$$- \sum_{x \in X_h} \sum_{a \in A} q^\star(x)\pi^\star(a|x) \left(b_t(x, a) + \left(1 + \frac{1}{H}\right) \mathbb{E}_{x' \sim P(\cdot|x,a)} \mathbb{E}_{a' \sim \pi_t(\cdot|x')} \left[B_t(x', a')\right]\right)$$

$$= \sum_{x \in X_h} \sum_{a \in A} q^\star(x)\pi_t(a|x)B_t(x, a) - \sum_{x \in X_{h+1}} \sum_{a \in A} q^\star(x)\pi_t(a|x)B_t(x, a)$$

$$-\sum_{x\in X_h}\sum_{a\in A}q^\star(x)\pi^\star(a|x)b_t(x,a)-\frac{1}{H}\sum_{x\in X_{h+1}}\sum_{a\in A}q^\star(x)\pi_t(a|x)B_t(x,a),$$

where the last step uses the fact $\sum_{x\in X_h}\sum_{a\in A}q^\star(x)\pi^\star(a|x)P(x'|x,a)=q^\star(x')$ (and then changes the notation $(x',a')$ to $(x,a)$). Now summing this over $h=0,1,\dots,H-1$ and $t=1,\dots,T$, and combining with $\text{TERM}_1$ and $\text{TERM}_2$, we get

$$\text{TERM}_1+\text{TERM}_2+\text{TERM}_3=\left(1+\frac{1}{H}\right)\sum_{t=1}^T\sum_a\pi_t(a|x_0)B_t(x_0,a).$$

Finally, we relate $\sum_a\pi_t(a|x_0)B_t(x_0,a)$ to $\widehat{V}^{\pi_t}(x_0;b_t)$. Below, we show by induction that for $x\in X_h$ and any $a$,

$$\sum_{a\in A}\pi_t(a|x)B_t(x,a)\le\left(1+\frac{1}{H}\right)^{H-h-1}\widehat{V}^{\pi_t}(x;b_t).$$

When $h=H-1$, $\sum_a\pi_t(a|x)B_t(x,a)=\sum_a\pi_t(a|x)b_t(x,a)=\widehat{V}^{\pi_t}(x;b_t)$. Suppose that the hypothesis holds for all $x\in X_h$. Then for any $x\in X_{h-1}$,

$$\sum_{a\in A}\pi_t(a|x)B_t(x,a)=\sum_a\pi_t(a|x)\left(b_t(x,a)+\left(1+\frac{1}{H}\right)\mathbb{E}_{x'\sim\widehat{P}(\cdot|x,a)}\mathbb{E}_{a'\sim\pi_t(\cdot|x')}\left[B_t(x',a')\right]\right)$$

$$\le\sum_a\pi_t(a|x)\left(b_t(x,a)+\left(1+\frac{1}{H}\right)^{H-h}\mathbb{E}_{x'\sim\widehat{P}(\cdot|x,a)}\left[\widehat{V}^{\pi_t}(x';b_t)\right]\right)$$

$$\text{(induction hypothesis)}$$

$$\le\left(1+\frac{1}{H}\right)^{H-h}\sum_a\pi_t(a|x)\left(b_t(x,a)+\mathbb{E}_{x'\sim\widehat{P}(\cdot|x,a)}\left[\widehat{V}^{\pi_t}(x';b_t)\right]\right)$$

$$(b_t(x,a)\ge0)$$

$$=\left(1+\frac{1}{H}\right)^{H-h}\widehat{V}^{\pi_t}(x;b_t),$$

finishing the induction. Applying the relation on $x=x_0$ and noticing that $\left(1+\frac{1}{H}\right)^H\le e<3$ finishing the proof. $\qquad\square$

Besides Lemma B.1, we also show Lemma B.2 below, which guarantees that Lemma 3.1 holds even if Eq. (4) and Eq. (5) only hold in expectation.

**Lemma B.2.** *Let $b_t(x,a)$ be a non-negative loss function that is fixed at the beginning of episode $t$, and let $\pi_t$ be fixed at the beginning of episode $t$. Let $B_t(x,a)$ be a randomized bonus function that satisfies the following for all $x,a$:*

$$\mathbb{E}_t\left[B_t(x,a)\right]=b_t(x,a)+\left(1+\frac{1}{H}\right)\mathbb{E}_{x'\sim P(\cdot|x,a)}\mathbb{E}_{a'\sim\pi_t(\cdot|x')}\mathbb{E}_t\left[B_t(x',a')\right]\qquad(14)$$

*with $B_t(x_H,a)\triangleq0$, and suppose that the following holds (simply taking expectations on Eq. (5)):*

$$\mathbb{E}\left[\sum_x q^\star(x)\sum_{t=1}^T\sum_a\left(\pi_t(a|x)-\pi^\star(a|x)\right)\left(Q_t^{\pi_t}(x,a)-B_t(x,a)\right)\right]$$

$$\le o(T)+\mathbb{E}\left[\sum_{t=1}^T V^{\pi^\star}(x_0;b_t)\right]+\frac{1}{H}\mathbb{E}\left[\sum_{t=1}^T\sum_{x,a}q^\star(x)\pi_t(a|x)B_t(x,a)\right].\qquad(15)$$

*Then*

$$\mathbb{E}\left[\text{Reg}\right]\le o(T)+3\mathbb{E}\left[\sum_{t=1}^T V^{\pi_t}(x_0;b_t)\right].$$

*Proof.* The proof of this lemma follows that of Lemma B.1 line-by-line (with $\widehat{P}=P$), except that we take expectations in all steps. $\qquad\square$

# C   Details Omitted in Section 4

In this section, we first discuss the implementation details of Algorithm 1 in Section C.1, then we give the complete proof of Theorem 4.1 in Section C.2.

## C.1   Implementation Details

The COMP-UOB procedure is the same as Algorithm 3 of [15], which shows how to efficiently compute an upper occupancy bound. We include the algorithm in Algorithm 4 for completeness. As Algorithm 1 also needs COMP-LOB, which computes a lower occupancy bound, we provide its complete pseudocode in Algorithm 5 as well.

Fix a state $x$. Define $f(\tilde{x})$ to be the maximum and minimum probability of visiting $x$ starting from state $\tilde{x}$ for COMP-UOB and COMP-LOB, respectively. Then the two algorithms almost have the same procedure to find $f(\tilde{x})$ by solving the optimization in Eq. (16) subject to $\widehat{P}$ in the confidence set $\mathcal{P}$ via a greedy approach in Algorithm 6. The difference is that COMP-UOB sets OPTIMIZE to be max while COMP-LOB sets OPTIMIZE to be min, and thus in Algorithm 6, $\{f(x)\}_{x \in X_k}$ is sorted in an ascending and a descending order, respectively.

Finally, we point out that the bonus function $B_t(s, a)$ defined in Eq. (9) can clearly also be computed using a greedy procedure similar to Algorithm 6. This concludes that the entire algorithm can be implemented efficiently.

$$f(\tilde{x}) = \sum_{a \in A} \pi_t(a|\tilde{x}) \left( \underset{\widehat{P}(\cdot|\tilde{x},a)}{\text{OPTIMIZE}} \sum_{x' \in X_{k(\tilde{x})+1}} \widehat{P}(x'|\tilde{x}, a) f(x') \right) \tag{16}$$

---

**Algorithm 4** COMP-UOB (Algorithm 3 of [15])

---

**Input:** a policy $\pi_t$, a state-action pair $(x, a)$ and a confidence set $\mathcal{P}$ of the form

$$\left\{ \widehat{P} : \left| \widehat{P}(x'|x, a) - \bar{P}(x'|x, a) \right| \leq \epsilon(x'|x, a), \ \forall (x, a, x') \right\}$$

**Initialize:** for all $\tilde{x} \in X_{k(x)}$, set $f(\tilde{x}) = \mathbb{1}\{\tilde{x} = x\}$.
**for** $k = k(x) - 1$ **to** $0$ **do**
    **for** $\forall \tilde{x} \in X_k$ **do**
        Compute $f(\tilde{x})$ based on :

$$f(\tilde{x}) = \sum_{a \in A} \pi_t(a|\tilde{x}) \cdot \text{GREEDY}\left( f, \bar{P}(\cdot|\tilde{x}, a), \epsilon(\cdot|\tilde{x}, a), \max \right)$$

**Return:** $\pi_t(a|x) f(x_0)$.

---

---

**Algorithm 5** COMP-LOB

---

**Input:** a policy $\pi_t$, a state-action pair $(x, a)$ and a confidence set $\mathcal{P}$ of the form

$$\left\{ \widehat{P} : \left| \widehat{P}(x'|x, a) - \bar{P}(x'|x, a) \right| \leq \epsilon(x'|x, a), \ \forall (x, a, x') \right\}$$

**Initialize:** for all $\tilde{x} \in X_{k(x)}$, set $f(\tilde{x}) = \mathbb{1}\{\tilde{x} = x\}$.
**for** $k = k(x) - 1$ **to** $0$ **do**
    **for** $\forall \tilde{x} \in X_k$ **do**
        Compute $f(\tilde{x})$ based on :

$$f(\tilde{x}) = \sum_{a \in A} \pi_t(a|\tilde{x}) \cdot \text{GREEDY}\left( f, \bar{P}(\cdot|\tilde{x}, a), \epsilon(\cdot|\tilde{x}, a), \min \right)$$

**Return:** $\pi_t(a|x) f(x_0)$.

---

**Algorithm 6** GREEDY

**Input:** $f : X \to [0,1]$, a distribution $\bar{p}$ over $n$ states of layer $k$ , positive numbers $\{\epsilon(x)\}_{x \in X_k}$, objective OPTIMIZE (max for COMP-UOB and min for COMP-LOB).
**Initialize:** $j^- = 1, j^+ = n$, sort $\{f(x)\}_{x \in X_k}$ and find $\sigma$ such that

$$f(\sigma(1)) \le f(\sigma(2)) \le \cdots \le f(\sigma(n))$$

for OPTIMIZE = max, and

$$f(\sigma(1)) \ge f(\sigma(2)) \ge \cdots \ge f(\sigma(n))$$

for OPTIMIZE = min.
**while** $j^- < j^+$ **do**
$\quad$ $x^- = \sigma(j^-), x^+ = \sigma(j^+)$
$\quad$ $\delta^- = \min\{\bar{p}(x^-), \epsilon(x^-)\}$
$\quad$ $\delta^+ = \min\{1 - \bar{p}(x^+), \epsilon(x^+)\}$
$\quad$ $\bar{p}(x^-) \leftarrow \bar{p}(x^-) - \min\{\delta^-, \delta^+\}$
$\quad$ $\bar{p}(x^+) \leftarrow \bar{p}(x^+) + \min\{\delta^-, \delta^+\}$
$\quad$ **if** $\delta_- \le \delta_+$ **then**
$\quad\quad$ $\epsilon(x^+) \leftarrow \epsilon(x^+) - \delta^-$
$\quad\quad$ $j^- \leftarrow j^- + 1$
$\quad$ **else**
$\quad\quad$ $\epsilon(x^-) \leftarrow \epsilon(x^-) - \delta^+$
$\quad\quad$ $j^+ \leftarrow j^+ - 1$
**Return:** $\sum_{j=1}^{n} \bar{p}(\sigma(j)) f(\sigma(j))$

## C.2 Omitted Proofs

To prove Theorem 4.1, as discussed in the analysis sketch of Section 4, we decompose the left-hand side of Eq. (5) as:

$$\sum_{t=1}^{T} \sum_{x} q^\star(x) \langle \pi_t(\cdot|x) - \pi^\star(\cdot|x), Q_t(x, \cdot) - B_t(x, \cdot) \rangle$$

$$= \underbrace{\sum_{t=1}^{T} \sum_{x} q^\star(x) \left\langle \pi_t(\cdot|x), Q_t(x, \cdot) - \widehat{Q}_t(x, \cdot) \right\rangle}_{\text{BIAS-1}} + \underbrace{\sum_{t=1}^{T} \sum_{x} q^\star(x) \left\langle \pi^\star(\cdot|x), \widehat{Q}_t(x, \cdot) - Q_t(x, \cdot) \right\rangle}_{\text{BIAS-2}}$$

$$+ \underbrace{\sum_{t=1}^{T} \sum_{x} q^\star(x) \left\langle \pi_t(\cdot|x) - \pi^\star(\cdot|x), \widehat{Q}_t(x, \cdot) - B_t(x, \cdot) \right\rangle}_{\text{REG-TERM}}. \tag{17}$$

We bound each term in a corresponding lemma. Specifically, We show a high probability bound of BIAS-1 in Lemma C.1, a high probability bound of BIAS-2 in Lemma C.2, and a high-probability bound of REG-TERM in Lemma C.3. Finally, we show how to combine all terms with the definition of $b_t$ in Theorem C.5, which is a restatement of Theorem 4.1.

**Lemma C.1** (BIAS-1). *With probability at least* $1 - 5\delta$,

$$\text{BIAS-1} \le \widetilde{\mathcal{O}} \left( \frac{H}{\eta} \right) + \sum_{t=1}^{T} \sum_{x,a} q^\star(x)\pi_t(a|x) \left( \frac{2\gamma H + H \left( \overline{q}_t(x, a) - \underline{q}_t(x, a) \right)}{\overline{q}_t(x, a) + \gamma} \right).$$

*Proof.* In the proof, we assume that $P \in \mathcal{P}_k$ for all $k$, with holds with probability at least $1 - 4\delta$ as already shown in [15, Lemma 2]. Under this event, $\underline{q}_t(x, a) \le q_t(x, a) \le \overline{q}_t(x, a)$ for all $t, x, a$.

Let $Y_t = \sum_{x \in X} q^\star(x) \left\langle \pi_t(\cdot|x), \widehat{Q}_t(x, \cdot) \right\rangle$. First, we decompose BIAS-1 as

$$\sum_{t=1}^{T} (\mathbb{E}_t[Y_t] - Y_t) + \left( \sum_{x} q^\star(x) \langle \pi_t(\cdot|x), Q_t(x, \cdot) \rangle - \mathbb{E}_t[Y_t] \right). \tag{18}$$

We will bound the first Martingale sequence using Freedman's inequality. Note that we have

$$
\begin{aligned}
\mathrm{Var}_t[Y_t] &\leq \mathbb{E}_t\left[\left(\sum_x q^\star(x)\left\langle \pi_t(\cdot|x), \widehat{Q}_t(x,\cdot)\right\rangle\right)^2\right] \\
&\leq \mathbb{E}_t\left[\left(\sum_{x,a} q^\star(x)\pi_t(a|x)\right)\left(\sum_{x,a} q^\star(x)\pi_t(a|x)\widehat{Q}_t(x,a)^2\right)\right] \quad \text{(Cauchy-Schwarz)} \\
&= H\sum_{x,a} q^\star(x)\pi_t(a|x)\frac{L_{t,h}^2\mathbb{E}_t[\mathbb{1}_t(x,a)]}{(\overline{q}_t(x,a)+\gamma)^2} \quad (\textstyle\sum_{x,a} q^\star(x)\pi_t(a|x) = H) \\
&\leq H\sum_{x,a} q^\star(x)\pi_t(a|x)\frac{q_t(x,a)H^2}{(\overline{q}_t(x,a)+\gamma)^2} \quad (L_{t,h}\leq H \text{ and } \mathbb{E}_t[\mathbb{1}_t(x,a)] = q_t(s,a)) \\
&\leq \sum_{x,a} q^\star(x)\pi_t(a|x)\frac{H^3}{\overline{q}_t(x,a)+\gamma} \quad (q_t(s,a)\leq \overline{q}_t(x,a))
\end{aligned}
$$

and $|Y_t|\leq H\sup_{x,a}|\widehat{Q}(x,a)|\leq \frac{H^2}{\gamma}$.

Moreover, for every $t$, the second term in Eq. (18) can be bounded as

$$
\begin{aligned}
&\sum_x q^\star(x)\left\langle \pi_t(\cdot|x), Q_t(x,\cdot)\right\rangle - \mathbb{E}_t\left[\sum_x q^\star(x)\left\langle \pi_t(\cdot|x), \widehat{Q}_t(x,\cdot)\right\rangle\right] \\
&= \sum_{x,a} q^\star(x)\pi_t(a|x)Q_t(x,a)\left(1 - \frac{q_t(x,a)}{\overline{q}_t(x,a)+\gamma}\right) \\
&\leq \sum_{x,a} q^\star(x)\pi_t(a|x)H\left(\frac{\overline{q}_t(x,a) - q_t(x,a) + \gamma}{\overline{q}_t(x,a)+\gamma}\right) \quad (Q_t(x,a)\leq H) \\
&\leq \sum_{x,a} q^\star(x)\pi_t(a|x)H\left(\frac{\overline{q}_t(x,a) - \underline{q}_t(x,a) + \gamma}{\overline{q}_t(x,a)+\gamma}\right). \quad (\underline{q}_t(x,a)\leq q_t(x,a))
\end{aligned}
$$

Combining them, and using Freedman's inequality (Lemma A.1), we have that with probability at least $1-5\delta$,

$$
\begin{aligned}
\textsc{Bias-1} &= \sum_{t=1}^T\sum_x q^\star(x)\left\langle \pi_t(\cdot|x), Q_t(x,\cdot) - \widehat{Q}_t(x,\cdot)\right\rangle \\
&\leq \sum_{t=1}^T\sum_{x,a} q^\star(x)\pi_t(a|x)H\left(\frac{\left(\overline{q}_t(x,a) - \underline{q}_t(x,a)\right) + \gamma}{\overline{q}_t(x,a)+\gamma}\right) \\
&\quad + \frac{\gamma}{H^2}\sum_{t=1}^T\sum_{x,a} q^\star(x)\pi_t(a|x)\frac{H^3}{\overline{q}_t(x,a)+\gamma} + \frac{H^2}{\gamma}\ln\frac{1}{\delta} \\
&\leq \widetilde{\mathcal{O}}\left(\frac{H}{\eta}\right) + \sum_{t=1}^T\sum_{x,a} q^\star(x)\pi_t(a|x)\left(\frac{2\gamma H + H\left(\overline{q}_t(x,a) - \underline{q}_t(x,a)\right)}{\overline{q}_t(x,a)+\gamma}\right),
\end{aligned}
$$

where we use $\gamma = 2\eta H$. $\qquad\square$

Next, we bound Bias-2.

**Lemma C.2** (Bias-2). *With probability at least $1-5\delta$, Bias-2 $\leq \widetilde{\mathcal{O}}\left(\frac{H}{\eta}\right)$.*

*Proof.* We invoke Lemma A.2 with $z_t(x,a) = q^\star(x)\pi^\star(a|x)Q_t(x,a)$ and $Z_t(x,a) = q^\star(x)\pi^\star(a|x)\left(\mathbb{1}_t(x,a)L_t(x,a) + (1 - \mathbb{1}_t(x,a))Q_t(x,a)\right)$. Then we get that with probability at

least $1 - \delta$ (recalling the definition $\widehat{Q}_t(x,a) = \frac{L_{t,h}}{\overline{q}_t(x,a)+\gamma}\mathbb{1}_t(x,a)$),

$$\sum_{t=1}^{T}\sum_{x,a} q^\star(x)\pi^\star(a|x)\left(\widehat{Q}_t(x,a) - \frac{q_t(x,a)}{\overline{q}_t(x,a)}Q_t(x,a)\right) \leq \frac{H^2}{2\gamma}\ln\frac{H}{\delta}, \tag{19}$$

Since with probability at least $1 - 4\delta$, $q_t(x,a) \leq \overline{q}_t(x,a)$ for all $t, x, a$ (by [15, Lemma 2]), Eq. (19) further implies that with probability at least $1 - 5\delta$,

$$\text{BIAS-2} = \sum_{t=1}^{T}\sum_{x,a} q^\star(x)\pi^\star(x,a)\left(\widehat{Q}_t(x,a) - Q_t(x,a)\right) \leq \frac{H^2}{2\gamma}\ln\frac{H}{\delta}.$$

Noting that $\gamma = 2\eta H$ finishes the proof. $\qquad\square$

We continue to bound REG-TERM.

**Lemma C.3** (REG-TERM). *With probability at least $1 - 5\delta$,*

$$\text{REG-TERM} \leq \widetilde{\mathcal{O}}\left(\frac{H}{\eta}\right) + \sum_{t=1}^{T}\sum_{x,a} q^\star(x)\pi_t(a|x)\left(\frac{\gamma H}{\overline{q}_t(x,a)+\gamma} + \frac{B_t(x,a)}{H}\right).$$

*Proof.* The algorithm runs individual exponential weight updates on each state with loss vectors $\widehat{Q}_t(x,\cdot) - B_t(x,\cdot)$, so we can apply standard results for exponential weight updates. Specifically, we can apply Lemma A.4 on each state $x$, and get

$$\sum_{t=1}^{T}\left\langle \pi_t(\cdot|x) - \pi^\star(\cdot|x), \widehat{Q}_t(x,\cdot) - B_t(x,\cdot)\right\rangle \leq \frac{\ln|A|}{\eta} + \eta\sum_{t=1}^{T}\sum_{a\in A}\pi_t(a|x)\left(\widehat{Q}_t(x,a) - B_t(x,a)\right)^2. \tag{20}$$

The condition required by Lemma A.4 (i.e., $\eta|\widehat{Q}_t(x,a) - B_t(x,a)| \leq 1$) is verified in Lemma C.4. Summing Eq. (20) over states with weights $q^\star(x)$, we get

$$\text{REG-TERM} \leq \frac{H\ln|A|}{\eta} + \eta\sum_{t=1}^{T}\sum_{x,a} q^\star(x)\pi_t(a|x)\left(\widehat{Q}_t(x,a) - B_t(x,a)\right)^2$$

$$\leq \frac{H\ln|A|}{\eta} + 2\eta\sum_{t=1}^{T}\sum_{x,a} q^\star(x)\pi_t(a|x)\widehat{Q}_t(x,a)^2 + 2\eta\sum_{t=1}^{T}\sum_{x,a} q^\star(x)\pi_t(a|x)B_t(x,a)^2. \tag{21}$$

Below, we focus on the last two terms on the right-hand side of Eq. (21). First, we have

$$2\eta\sum_{t=1}^{T}\sum_{x,a} q^\star(x)\pi_t(a|x)\widehat{Q}_t(x,a)^2 \leq 2\eta\sum_{t=1}^{T}\sum_{x,a} q^\star(x)\pi_t(a|x)\frac{H^2\mathbb{1}_t(x,a)}{(\overline{q}_t(x,a)+\gamma)^2}$$

$$= 2\eta H^2\sum_{t=1}^{T}\sum_{x,a}\frac{q^\star(x)\pi_t(a|x)}{\overline{q}_t(x,a)+\gamma}\cdot\frac{\mathbb{1}_t(x,a)}{\overline{q}_t(x,a)+\gamma}$$

$$\leq 2\eta H^2\sum_{t=1}^{T}\sum_{x,a}\frac{q^\star(x)\pi_t(a|x)}{\overline{q}_t(x,a)+\gamma}\cdot\frac{q_t(x,a)}{\overline{q}_t(x,a)} + 2\eta H^2\times\frac{\frac{H}{\gamma}\ln\frac{H}{\delta}}{2\gamma}$$

$$\leq \frac{H}{4\eta}\ln\frac{H}{\delta} + \sum_{t=1}^{T}\sum_{x,a} q^\star(x)\pi_t(a|x)\frac{\gamma H}{\overline{q}_t(x,a)+\gamma},$$

where the third step happens with probability at least $1 - \delta$ by Lemma A.2 with $z_t(x,a) = Z_t(x,a) = \frac{q^\star(x)\pi_t(a|x)}{\overline{q}_t(x,a)+\gamma} \leq \frac{1}{\gamma}$, and the last step uses $\gamma = 2\eta H$ and $q_t(x,a) \leq \overline{q}_t(x,a)$ (which happens with probability at least $1 - 4\delta$). For the second term in Eq. (21), note that

$$2\eta\sum_{t=1}^{T}\sum_{a\in A}\pi_t(a|x)B_t(x,a)^2 \leq \frac{1}{H}\sum_{t=1}^{T}\sum_{a\in A}\pi_t(a|x)B_t(x,a)$$

due to the fact $\eta B_t(x,a) \le \frac{1}{2H}$ by Lemma C.4. Combining everything finishes the proof. $\qquad\square$

In Lemma C.3, as required by Lemma A.4, we control the magnitude of $\eta\widehat{Q}_t(x,a)$ and $\eta B_t(x,a)$ by setting $\gamma$ and $\eta$ properly, shown in the following technical lemma.

**Lemma C.4.** $\eta\widehat{Q}_t(x,a) \le \frac{1}{2}$ and $\eta B_t(x,a) \le \frac{1}{2H}$.

*Proof.* Recall that $\gamma = 2\eta H$ and $\eta \le \frac{1}{24H^3}$. Thus,

$$\eta\widehat{Q}_t(x,a) \le \frac{\eta H}{\gamma} = \frac{\eta H}{2\eta H} = \frac{1}{2},$$

$$\eta b_t(x,a) = \frac{3\eta\gamma H + \eta H(\overline{q}_t(x,a) - \underline{q}_t(x,a))}{\overline{q}_t(x,a) + \gamma} \le 3\eta H + \eta H \le \frac{1}{6H^2}.$$

By the definition of $B_t(x,a)$ in Eq. (9), we have

$$\eta B_t(x,a) \le H\left(1 + \frac{1}{H}\right)^H \eta \sup_{x',a'} b_t(x',a') \le 3H \times \frac{1}{6H^2} = \frac{1}{2H}.$$

This finishes the proof. $\qquad\square$

Now we are ready to prove Theorem 4.1. For convenience, we state the theorem again here and show the proof.

**Theorem C.5.** *Algorithm 1 ensures that with probability $1 - \mathcal{O}(\delta)$, $\mathrm{Reg} = \widetilde{\mathcal{O}}\left(|X|H^2\sqrt{AT} + H^4\right)$.*

*Proof.* Combining BIAS-1, BIAS-2, REG-TERM, we get that with probability at least $1 - \mathcal{O}(\delta)$,

$$\mathrm{BIAS\text{-}1} + \mathrm{BIAS\text{-}2} + \mathrm{REG\text{-}TERM}$$

$$\le \widetilde{\mathcal{O}}\left(\frac{H}{\eta}\right) + \sum_{t=1}^{T}\sum_{x,a} q^\star(x)\pi_t(a|x)\left(\frac{3\gamma H + H(\overline{q}_t(x,a) - \underline{q}_t(x,a))}{\overline{q}_t(x,a) + \gamma} + \frac{1}{H}B_t(x,a)\right)$$

$$= \widetilde{\mathcal{O}}\left(\frac{H}{\eta}\right) + \sum_{t=1}^{T}\sum_{x,a} q^\star(x)\pi^\star(a|x)b_t(x,a) + \frac{1}{H}\sum_{t=1}^{T}\sum_{x,a} q^\star(x)\pi_t(a|x)B_t(x,a),$$

which is of the form specified in Eq. (5). By the definition of $B_t(x,a)$ in Eq. (9), we see that Eq. (13) also holds with probability at least $1 - \mathcal{O}(\delta)$ for all $t, x, a$.

Therefore, by Lemma B.1, we can bound the regret as (let $\widehat{P}_t$ be the optimistic transition function chosen in Eq. (9) at episode $t$)

$$\mathrm{Reg} = \widetilde{\mathcal{O}}\left(\frac{H}{\eta} + \sum_{t=1}^{T}\sum_{x,a} q^{\widehat{P}_t,\pi_t}(x,a)b_t(x,a)\right)$$

$$= \widetilde{\mathcal{O}}\left(\frac{H}{\eta} + \sum_{t=1}^{T}\sum_{x,a} q^{\widehat{P}_t,\pi_t}(x,a)\frac{H(\overline{q}_t(x,a) - \underline{q}_t(x,a)) + \gamma H}{\overline{q}_t(x,a) + \gamma}\right)$$

$$= \widetilde{\mathcal{O}}\left(\frac{H}{\eta} + \sum_{t=1}^{T}\sum_{x,a}\left(H(\overline{q}_t(x,a) - \underline{q}_t(x,a)) + \eta H^2\right)\right)$$

$$\qquad\qquad\qquad\qquad\qquad (q^{\widehat{P}_t,\pi_t}(x,a) \le \overline{q}_t(x,a) \text{ and } \gamma = 2\eta H)$$

$$\le \widetilde{\mathcal{O}}\left(\frac{H}{\eta} + |X|H^2\sqrt{AT} + \eta|X||A|H^2 T\right),$$

where the last inequality is due to [15, Lemma 4]. Plugging in the specified value for $\eta$, the regret can be further upper bounded by $\widetilde{\mathcal{O}}\left(|X|H^2\sqrt{AT} + H^4\right)$. $\qquad\square$

# D   Details Omitted in Section 5

In this section, our goal is to analyze Algorithm 2 and prove Theorem 5.1. Before conducting regret analysis, we first analyze the GEOMETRICRESAMPLING algorithm in Appendix D.1, which mostly follows [24].

## D.1   GEOMETRICRESAMPLING and Its Analysis

The GEOMETRICRESAMPLING algorithm is shown in Algorithm 7, which is almost the same as that in [24] except that we repeat the same procedure for $M$ times and average the outputs (see the extra outer loop). This extra step is added to deal with some technical difficulties in the analysis.

---

**Algorithm 7** GEOMETRICRESAMPLING$(t, M, N, \gamma)$

---

Let $c = \frac{1}{2}$.
**for** $m = 1, \ldots, M$ **do**
    **for** $n = 1, \ldots, N$ **do**
        Generate path $(x_{n,0}, a_{n,0}), \ldots, (x_{n,H-1}, a_{n,H-1})$ using policy $\pi_t$ and the simulator.
        For all $h$, compute $Y_{n,h} = \gamma I + \phi(x_{n,h}, a_{n,h})\phi(x_{n,h}, a_{n,h})^\top$.
        For all $h$, compute $Z_{n,h} = \Pi_{j=1}^n (I - cY_{j,h})$.
    For all $h$, set $\widehat{\Sigma}_{t,h}^{+(m)} = cI + c\sum_{n=1}^N Z_{n,h}$.
For all $h$, set $\widehat{\Sigma}_{t,h}^+ = \frac{1}{M}\sum_{m=1}^M \widehat{\Sigma}_{t,h}^{+(m)}$.
**return** $\widehat{\Sigma}_{t,h}^+$ for all $h = 0, \ldots, H-1$.

---

**Lemma D.1.** *Let* $M = \left\lceil \frac{24\ln(dHT)}{\epsilon^2\gamma^2} \right\rceil$, $N = \left\lceil \frac{2}{\gamma}\ln\frac{1}{\epsilon\gamma} \right\rceil$ *for some* $\epsilon, \gamma > 0$. *Then* GEOMETRICRESAMPLING *(Algorithm 7) with input* $(t, M, N, \gamma)$ *ensures the following for all* $h$:

$$\left\| \widehat{\Sigma}_{t,h}^+ \right\|_{\text{op}} \leq \frac{1}{\gamma}. \tag{22}$$

$$\left\| \mathbb{E}_t\left[\widehat{\Sigma}_{t,h}^+\right] - (\gamma I + \Sigma_{t,h})^{-1} \right\|_{\text{op}} \leq \epsilon, \tag{23}$$

$$\left\| \widehat{\Sigma}_{t,h}^+ - (\gamma I + \Sigma_{t,h})^{-1} \right\|_{\text{op}} \leq 2\epsilon, \tag{24}$$

$$\left\| \widehat{\Sigma}_{t,h}^+ \Sigma_{t,h} \right\|_{\text{op}} \leq 1 + 2\epsilon, \tag{25}$$

*where* $\|\cdot\|_{\text{op}}$ *represents the spectral norm and the last two properties Eq. (24) and Eq. (25) hold with probability at least* $1 - \frac{1}{T^3}$.

*Proof.* To prove Eq. (22), notice that each one of $\widehat{\Sigma}_{t,h}^{+(m)}$, $m = 1, \ldots, M$, is a sum of $N + 1$ terms. Furthermore, the $n$-th term of them ($cZ_{n,h}$ in Algorithm 7) has an operator norm upper bounded by $c(1 - c\gamma)^n$. Therefore,

$$\left\| \widehat{\Sigma}_{t,h}^{+(m)} \right\|_{\text{op}} \leq \sum_{n=0}^N c(1 - c\gamma)^n \leq \frac{1}{\gamma}. \tag{26}$$

Since $\widehat{\Sigma}_{t,h}^+$ is an average of $\widehat{\Sigma}_{t,h}^{+(m)}$, this implies Eq. (22).

To show Eq. (23), observe that $\mathbb{E}_t[Y_{n,h}] = \gamma I + \Sigma_{t,h}$ and $\{Y_{n,h}\}_{n=1}^N$ are independent. Therefore, we a have

$$\mathbb{E}_t\left[\widehat{\Sigma}_{t,h}^+\right] = \mathbb{E}_t\left[\widehat{\Sigma}_{t,h}^{+(m)}\right] = cI + c\sum_{i=1}^N (I - c(\gamma I + \Sigma_{t,h}))^i$$

$$= (\gamma I + \Sigma_{t,h})^{-1}\left(I - (I - c(\gamma I + \Sigma_{t,h}))^{N+1}\right)$$

where the last step uses the formula: $\left(I + \sum_{i=1}^N A^i\right) = (I-A)^{-1}(I - A^{N+1})$ with $A = I - c(\gamma I + \Sigma_{t,h})$. Thus,

$$\left\| \mathbb{E}_t \left[ \widehat{\Sigma}_{t,h}^+ \right] - (\gamma I + \Sigma_{t,h})^{-1} \right\|_{\mathrm{op}} = \left\| (\gamma I + \Sigma_{t,h})^{-1} \left( I - c\left( \gamma I + \Sigma_{t,h} \right) \right)^{N+1} \right\|_{\mathrm{op}}$$

$$\leq \frac{(1 - c\gamma)^{N+1}}{\gamma} \leq \frac{e^{-(N+1)c\gamma}}{\gamma} \leq \epsilon,$$

where the first inequality is by $0 \prec I - c(\gamma I + I) \preceq I - c(\gamma I + \Sigma_{t,h}) \preceq I - c\gamma I$, and the last inequality is by our choice of $N$ and that $c = \frac{1}{2}$.

To show Eq. (24), we only further need

$$\left\| \widehat{\Sigma}_{t,h}^+ - \mathbb{E}_t \left[ \widehat{\Sigma}_{t,h}^+ \right] \right\|_{\mathrm{op}} \leq \epsilon$$

and combine it with Eq. (23). This can be shown by applying Lemma A.3 with $X_k = \widehat{\Sigma}_{t,h}^{+(k)} - \mathbb{E}_t \left[ \widehat{\Sigma}_{t,h}^{+(k)} \right]$, $A_k = \frac{1}{\gamma} I$ (recall Eq. (26) and thus $X_k^2 \preceq A_k^2$), $\sigma = \frac{1}{\gamma}$, $\tau = \epsilon$, and $n = M$. This gives the following statement: the event $\left\| \widehat{\Sigma}_{t,h}^+ - \mathbb{E}_t \left[ \widehat{\Sigma}_{t,h}^+ \right] \right\|_{\mathrm{op}} > \epsilon$ holds with probability less than

$$d \exp\left( -M \times \epsilon^2 \times \frac{1}{8} \times \gamma^2 \right) \leq \frac{1}{d^2 H^3 T^3} \leq \frac{1}{H T^3}$$

by our choice of $M$. The conclusion follows by a union bound over $h$.

To prove Eq. (25), observe that with Eq. (24), we have

$$\left\| \widehat{\Sigma}_{t,h}^+ \Sigma_{t,h} \right\|_{\mathrm{op}} \leq \left\| (\gamma I + \Sigma_{t,h})^{-1} \Sigma_{t,h} \right\|_{\mathrm{op}} + \left\| \left( \widehat{\Sigma}_{t,h}^+ - (\gamma I + \Sigma_{t,h})^{-1} \right) \Sigma_{t,h} \right\|_{\mathrm{op}} \leq 1 + 2\epsilon$$

since $\| \Sigma_{t,h} \|_{\mathrm{op}} \leq 1$. $\qquad\square$

## D.2 Regret Analysis

In the analysis, we require that $\pi_t(a|x)$ and $B_t(x,a)$ be defined for all $x, a, t$, but in Algorithm 2, they are only explicitly defined if the learner has ever visited state $x$. Below, we construct a virtual process that is equivalent to Algorithm 2, but with all $\pi_t(a|x)$ and $B_t(x,a)$ well-defined.

Imagine a virtual process where at the end of episode $t$ (a moment when $\widehat{\Sigma}_t^+$ has been defined), BONUS$(t,x,a)$ is called once for every $(x,a)$, in an order from layer $H-1$ to layer 0. Observe that within BONUS$(t,x,a)$, other BONUS$(t',x',a')$ might be called, but either $t' < t$, or $x'$ is in a later layer. Therefore, in this virtual process, every recursive call will soon be returned in the third line of Algorithm 3 because they have been called previously and the values of them are already determined. Given that BONUS$(t,x,a)$ are all called once, at the beginning of episode $t+1$, $\pi_{t+1}$ will be well-defined for all states since it only depends on BONUS$(t',x',a')$ with $t' \leq t$ and other quantities that are well-defined before episode $t+1$.

Comparing the virtual process and the real process, we see that the virtual process calculates all entries of BONUS$(t,x,a)$, while the real process only calculates a subset of them that are necessary for constructing $\pi_t$ and $\widehat{\Sigma}_t^+$. However, they define exactly the same policies as long as the random seeds we use for each entry of BONUS$(t,x,a)$ are the same for both processes. Therefore, we can define $B_t(x,a)$ unambiguously as the value returned by BONUS$(t,x,a)$ in the virtual process, and $\pi_t(a|x)$ as shown in (11) with BONUS$(\tau,x,a)$ replaced by $B_\tau(x,a)$.

Now, we follow the exactly same regret decomposition as described in Section 4 (see also Eq. (17)), with the new definition of $\widehat{Q}_t(x,a) \triangleq \phi(x,a)^\top \widehat{\theta}_{t,h}$ (for $x \in X_h$) and $B_t(x,a)$ described above, and then bound $\mathbb{E}[\text{BIAS-1} + \text{BIAS-2}]$ and $\mathbb{E}[\text{REG-TERM}]$ in Lemma D.2 and Lemma D.3 respectively.

**Lemma D.2.** *If $\beta \leq H$, then $\mathbb{E}[\text{BIAS-1} + \text{BIAS-2}]$ is upper bounded by*

$$\frac{\beta}{4} \mathbb{E}\left[ \sum_{t=1}^T \sum_{h=0}^{H-1} \sum_{(x,a) \in X_h \times A} q^\star(x) \left( \pi_t(a|x) + \pi^\star(a|x) \right) \| \phi(x,a) \|_{\widehat{\Sigma}_{t,h}^+}^2 \right] + \mathcal{O}\left( \frac{\gamma d H^3 T}{\beta} + \epsilon H^2 T \right).$$

*Proof of Lemma D.2.* Consider a specific $(t, x, a)$. Let $h$ be such that $x \in X_h$. Then we proceed as

$$\mathbb{E}_t \left[ Q_t^{\pi_t}(x, a) - \widehat{Q}_t(x, a) \right]$$

$$= \phi(x, a)^\top \left( \theta_{t,h}^{\pi_t} - \mathbb{E}_t \left[ \widehat{\theta}_{t,h} \right] \right)$$

$$= \phi(x, a)^\top \left( \theta_{t,h}^{\pi_t} - \mathbb{E}_t \left[ \widehat{\Sigma}_{t,h}^+ \right] \mathbb{E}_t \left[ \phi(x_{t,h}, a_{t,h}) L_{t,h} \right] \right) \qquad \text{(definition of } \widehat{\theta}_{t,h})$$

$$= \phi(x, a)^\top \left( \theta_{t,h}^{\pi_t} - (\gamma I + \Sigma_{t,h})^{-1} \mathbb{E}_t \left[ \phi(x_{t,h}, a_{t,h}) L_{t,h} \right] \right) + \mathcal{O}(\epsilon H)$$
$$\text{(by Eq. (23) of Lemma D.1 and that } \|\phi(x, a)\| \leq 1 \text{ for all } x, a \text{ and } L_{t,h} \leq H)$$

$$= \phi(x, a)^\top \left( \theta_{t,h}^{\pi_t} - (\gamma I + \Sigma_{t,h})^{-1} \Sigma_{t,h} \theta_{t,h}^{\pi_t} \right) + \mathcal{O}(\epsilon H) \qquad (\mathbb{E}[L_{t,h}] = \phi(x_{t,h}, a_{t,h})^\top \theta_{t,h}^{\pi_t})$$

$$= \gamma \phi(x, a)^\top (\gamma I + \Sigma_{t,h})^{-1} \theta_{t,h}^{\pi_t} + \mathcal{O}(\epsilon H) \qquad (\theta_{t,h}^{\pi_t} = (\gamma I + \Sigma_{t,h})^{-1} (\gamma I + \Sigma_{t,h}) \theta_{t,h}^{\pi_t})$$

$$\leq \gamma \|\phi(x, a)\|_{(\gamma I + \Sigma_{t,h})^{-1}}^2 \|\theta_{t,h}^{\pi_t}\|_{(\gamma I + \Sigma_{t,h})^{-1}}^2 + \mathcal{O}(\epsilon H) \qquad \text{(Cauchy-Schwarz inequality)}$$

$$\leq \frac{\beta}{4} \|\phi(x, a)\|_{(\gamma I + \Sigma_{t,h})^{-1}}^2 + \frac{\gamma^2}{\beta} \|\theta_{t,h}^{\pi_t}\|_{(\gamma I + \Sigma_{t,h})^{-1}}^2 + \mathcal{O}(\epsilon H) \qquad \text{(AM-GM inequality)}$$

$$\leq \frac{\beta}{4} \mathbb{E}_t \left[ \|\phi(x, a)\|_{\widehat{\Sigma}_{t,h}^+}^2 \right] + \frac{\gamma d H^2}{\beta} + \mathcal{O}\left( \epsilon (H + \beta) \right) \qquad (27)$$

where in the last inequality we use Eq. (23) again and also $\|\theta_{t,h}^{\pi}\|^2 \leq d H^2$ according to Assumption 1. Summing the above over $t, x, a$ with weights $q^\star(x) \pi_t(a|x)$, and taking expectation, we get

$$\mathbb{E}[\text{BIAS-1}] \leq \frac{\beta}{4} \mathbb{E} \left[ \sum_{t=1}^T \sum_{h=0}^{H-1} \sum_{(x,a) \in X_h \times A} q^\star(x) \pi_t(a|x) \|\phi(x, a)\|_{\widehat{\Sigma}_{t,h}^+}^2 \right] + \mathcal{O}\left( \frac{\gamma d H^3 T}{\beta} + \epsilon H^2 T \right).$$
$$\text{(using } \beta \leq H)$$

By the same argument, we can show that $\mathbb{E}_t[\widehat{Q}_t(x, a) - Q_t^{\pi_t}(x, a)]$ is also upper bounded by the right-hand side of Eq. (27), and thus

$$\mathbb{E}[\text{BIAS-2}] \leq \frac{\beta}{4} \mathbb{E} \left[ \sum_{t=1}^T \sum_{h=0}^{H-1} \sum_{(x,a) \in X_h \times A} q^\star(x) \pi^\star(a|x) \|\phi(x, a)\|_{\widehat{\Sigma}_{t,h}^+}^2 \right] + \mathcal{O}\left( \frac{\gamma d H^3 T}{\beta} + \epsilon H^2 T \right).$$

Summing them up finishes the proof. $\qquad \square$

**Lemma D.3.** *If $\eta \beta \leq \frac{\gamma}{12 H^2}$ and $\eta \leq \frac{\gamma}{2H}$, then $\mathbb{E}[\text{REG-TERM}]$ is upper bounded by*

$$\frac{H \ln |A|}{\eta} + 2 \eta H^2 \mathbb{E} \left[ \sum_{t=1}^T \sum_{h=0}^{H-1} \sum_{(x,a) \in X_h \times A} q^\star(x) \pi_t(a|x) \|\phi(x, a)\|_{\widehat{\Sigma}_{t,h}^+}^2 \right]$$

$$+ \frac{1}{H} \mathbb{E} \left[ \sum_{t=1}^T \sum_{x,a} q^\star(x) \pi_t(a|x) B_t(x, a) \right] + \mathcal{O}\left( \eta \epsilon H^3 T + \frac{\eta H^3}{\gamma^2 T^2} \right).$$

*Proof of Lemma D.3.* Again, we will apply the regret bound of the exponential weight algorithm Lemma A.4 to each state. We start by checking the required condition: $\eta |\phi(x, a)^\top \widehat{\theta}_{\tau,h} - B_t(x, a)| \leq 1$. This can be seen by that

$$\eta \left| \phi(x, a)^\top \widehat{\theta}_{\tau,h} \right| = \eta \left| \phi(x, a)^\top \widehat{\Sigma}_{t,h}^+ \phi(x_{t,h}, a_{t,h}) L_{t,h} \right|$$

$$\leq \eta \times \left\| \widehat{\Sigma}_{t,h}^+ \right\|_{\text{op}} \times L_{t,h} \leq \frac{\eta H}{\gamma} \leq \frac{1}{2}, \qquad \text{(Eq. (22) and the condition } \eta \leq \frac{\gamma}{2H})$$

and that by the definition of BONUS$(t, x, a)$, we have

$$\eta B_t(x, a) \leq \eta \times H \left( 1 + \frac{1}{H} \right)^H \times 2 \beta \sup_{x,a,h} \|\phi(x, a)\|_{\widehat{\Sigma}_{t,h}^+}^2 \leq \frac{6 \eta \beta H}{\gamma} \leq \frac{1}{2H}, \qquad (28)$$

where the last inequality is by Eq. (22) again and the condition $\eta\beta \leq \frac{\gamma}{12H^2}$.

Thus, by Lemma A.4, we have for any $x$,

$$\mathbb{E}\left[\sum_{t=1}^{T}\sum_{a}\left(\pi_t(a|x) - \pi^\star(a|x)\right)\widehat{Q}_t(x,a)\right]$$

$$\leq \frac{\ln|A|}{\eta} + 2\eta\mathbb{E}\left[\sum_{t=1}^{T}\sum_{a}\pi_t(a|x)\widehat{Q}_t(x,a)^2\right] + 2\eta\mathbb{E}\left[\sum_{t=1}^{T}\sum_{a}\pi_t(a|x)B_t(x,a)^2\right]. \qquad (29)$$

The last term in Eq. (29) can be upper bounded by $\mathbb{E}\left[\frac{1}{H}\sum_{t=1}^{T}\sum_{a}\pi_t(a|x)B_t(x,a)\right]$ because $\eta B_t(x,a) \leq \frac{1}{2H}$ as we verified in Eq. (28). To bound the second term in Eq. (29), we use the following: for $(x,a) \in X_h \times A$,

$$\mathbb{E}_t\left[\widehat{Q}_t(x,a)^2\right] \leq H^2 \mathbb{E}_t\left[\phi(x,a)^\top \widehat{\Sigma}_{t,h}^+ \phi(x_{t,h}, a_{t,h})\phi(x_{t,h}, a_{t,h})^\top \widehat{\Sigma}_{t,h}^+ \phi(x,a)\right]$$

$$= H^2 \mathbb{E}_t\left[\phi(x,a)^\top \widehat{\Sigma}_{t,h}^+ \Sigma_{t,h}\widehat{\Sigma}_{t,h}^+ \phi(x,a)\right]$$

$$\leq H^2 \mathbb{E}_t\left[\phi(x,a)^\top \widehat{\Sigma}_{t,h}^+ \Sigma_{t,h}\left(\gamma I + \Sigma_{t,h}\right)^{-1}\phi(x,a)\right] + \mathcal{O}\left(\epsilon H^2 + \frac{H^2}{\gamma^2 T^3}\right) \qquad (*)$$

$$\leq H^2\phi(x,a)^\top \left(\gamma I + \Sigma_{t,h}\right)^{-1}\Sigma_{t,h}\left(\gamma I + \Sigma_{t,h}\right)^{-1}\phi(x,a) + \mathcal{O}\left(\epsilon H^2 + \frac{H^2}{\gamma^2 T^3}\right)$$
$$\text{(by Eq. (23))}$$

$$\leq H^2\phi(x,a)^\top \left(\gamma I + \Sigma_{t,h}\right)^{-1}\phi(x,a) + \mathcal{O}\left(\epsilon H^2 + \frac{H^2}{\gamma^2 T^3}\right)$$

$$\leq H^2 \mathbb{E}_t\left[\phi(x,a)^\top \widehat{\Sigma}_{t,h}^+ \phi(x,a)\right] + \mathcal{O}\left(\epsilon H^2 + \frac{H^2}{\gamma^2 T^3}\right) \qquad \text{(by Eq. (23) again)}$$

$$= H^2 \mathbb{E}_t\left[\|\phi(x,a)\|_{\widehat{\Sigma}_{t,h}^+}^2\right] + \mathcal{O}\left(\epsilon H^2 + \frac{H^2}{\gamma^2 T^3}\right)$$

where $(*)$ is because by Eq. (24) and Eq. (25), $\|(\gamma I + \Sigma_{t,h})^{-1} - \widehat{\Sigma}_{t,h}^+\|_{\text{op}} \leq 2\epsilon$ and $\|\widehat{\Sigma}_{t,h}^+ \Sigma_{t,h}\|_{\text{op}} \leq 1 + 2\epsilon$ hold with probability $1 - \frac{1}{T^3}$; for the remaining probability, we upper bound $H^2\phi(x,a)^\top \widehat{\Sigma}_{t,h}^+ \Sigma_{t,h}\widehat{\Sigma}_{t,h}^+ \phi(x,a)$ by $\frac{H^2}{\gamma^2}$. Combining them with Eq. (29) and summing over states with weights $q^\star(x)$ finishes the proof. $\qquad\square$

With Lemma D.2 and Lemma D.3, we can now prove Theorem 5.1.

*Proof of Theorem 5.1.* Combining Lemma D.2 and Lemma D.3, we get (under the required conditions of the parameters):

$$\mathbb{E}\left[\text{BIAS-1} + \text{BIAS-2} + \text{REG-TERM}\right]$$

$$\leq \mathcal{O}\left(\frac{H\ln|A|}{\eta} + \frac{\gamma d H^3 T}{\beta} + \epsilon H^2 T + \eta\epsilon H^3 T + \frac{\eta H^3}{\gamma^2 T^2}\right)$$

$$+ \left(2\eta H^2 + \frac{\beta}{4}\right)\mathbb{E}\left[\sum_{t=1}^{T}\sum_{h=0}^{H-1}\sum_{(x,a)\in X_h\times A} q^\star(x)\left(\pi_t(a|x) + \pi^\star(a|x)\right)\|\phi(x,a)\|_{\widehat{\Sigma}_{t,h}^+}^2\right]$$

$$+ \frac{1}{H}\mathbb{E}\left[\sum_{t=1}^{T}\sum_{x,a} q^\star(x)\pi_t(a|x)B_t(x,a)\right].$$

We see that Eq. (15) is satisfied in expectation as long as we have $2\eta H^2 + \frac{\beta}{4} \leq \beta$ and define $b_t(x,a) \triangleq \beta\|\phi(x,a)\|_{\widehat{\Sigma}_{t,h}^+}^2 + \beta\sum_{a'}\pi_t(a'|x)\|\phi(x,a')\|_{\widehat{\Sigma}_{t,h}^+}^2$ (for $x \in X_h$). By the definition of Algorithm 3, Eq. (14) is also satisfied with this choice of $b_t(x,a)$. Therefore, we can apply

[Lemma B.2](#) to obtain a regret bound. To simply the presentation, we first pick $\epsilon = \frac{1}{H^3 T}$ so that all $\epsilon$-related terms become $\mathcal{O}(1)$. Then we have

$$\mathbb{E}[\text{Reg}]$$

$$= \widetilde{\mathcal{O}}\left( \frac{H}{\eta} + \frac{\gamma d H^3 T}{\beta} + \frac{\eta H^3}{\gamma^2 T^2} + \mathbb{E}\left[ \sum_{t=1}^{T} \sum_{x,a} q_t(x,a) b_t(x,a) \right] \right)$$

$$= \widetilde{\mathcal{O}}\left( \frac{H}{\eta} + \frac{\gamma d H^3 T}{\beta} + \frac{\eta H^3}{\gamma^2 T^2} + \beta\mathbb{E}\left[ \sum_{t=1}^{T} \sum_{h} \sum_{(x,a)\in X_h \times A} q_t(x,a) \|\phi(x,a)\|^2_{\widehat{\Sigma}^+_{t,h}} \right] \right)$$

$$= \widetilde{\mathcal{O}}\left( \frac{H}{\eta} + \frac{\gamma d H^3 T}{\beta} + \frac{\eta H^3}{\gamma^2 T^2} + \beta\mathbb{E}\left[ \sum_{t=1}^{T} \sum_{h} \sum_{(x,a)\in X_h \times A} q_t(x,a) \|\phi(x,a)\|^2_{(\gamma I + \Sigma_{t,h})^{-1}} \right] \right)$$

$$\text{([Eq. (23)](#) and } \beta \le H)$$

$$= \widetilde{\mathcal{O}}\left( \frac{H}{\eta} + \frac{\gamma d H^3 T}{\beta} + \frac{\eta H^3}{\gamma^2 T^2} + \beta d H T \right),$$

where the last step uses the fact

$$\mathbb{E}_t\left[ \sum_{h} \sum_{(x,a)\in X_h \times A} q_t(x,a) \|\phi(x,a)\|^2_{(\gamma I + \Sigma_{t,h})^{-1}} \right] \le \mathbb{E}_t\left[ \sum_{h} \sum_{(x,a)\in X_h \times A} q_t(x,a) \|\phi(x,a)\|^2_{\Sigma_{t,h}^{-1}} \right]$$

$$= \sum_{h} \left\langle \Sigma_{t,h}, \Sigma_{t,h}^{-1} \right\rangle = dH. \tag{30}$$

Finally, choosing the parameters under the specified constraints as:

$$\gamma = (dT)^{-\frac{2}{3}}, \qquad \beta = H(dT)^{-\frac{1}{3}}, \qquad \epsilon = \frac{1}{H^3 T},$$

$$\eta = \min\left\{ \frac{\gamma}{2H}, \frac{3\beta}{8H^2}, \frac{\gamma}{12\beta H^2} \right\},$$

we further bound the regret by $\widetilde{\mathcal{O}}\left( H^2 (dT)^{\frac{2}{3}} + H^4 (dT)^{\frac{1}{3}} \right)$. $\qquad\qquad\square$

# E  Details for Linear-$Q$ with a Simulator and an Exploratory Policy

The main algorithm ([Algorithm 8](#)) follows the same idea as [Algorithm 2](#). The main difference is that we can leverage $\pi_0$ to perform exploration. To do so, in Step 1 of the algorithm, we draw a Bernoulli random variable $Y_t \sim \text{BERNOULLI}(\delta_e)$ (for some $\delta_e \in (0,1)$) to indicate whether in this round the learner should use $\pi_0$. If $Y_t$ is 1, then the learner further randomly draw $h_t^*$ from $0, \ldots, H-1$. Then she walks from $x_0$ to layer $h_t^*$ using $\pi_0$, and then continues with $\pi_t$ to the end. In this way, the learner can explicitly explore the state space on every layer, which facilitates estimating $\theta_{t,h}^{\pi_t}$ with less bias.

Because we mix the exploration into the policy, we perform a slightly different procedure GEOMETRICRESAMPLING-MIXTURE in Step 2, which does not incorporate the $\gamma$ parameter as in GEOMETRICRESAMPLING. Instead, it will estimate the inverse of $\Sigma_{t,h}^{\text{mix}} = (1-\delta_e)\Sigma_{t,h} + \delta_e \Sigma_h^{\pi_0}$ where $\Sigma_{t,h}$ is the covariance matrix under $\pi_t$ and $\Sigma_h^{\pi_0}$ is the covariance matrix under $\pi_0$.

The new construction of $\widehat{\theta}_{t,h}$ in Step 3 makes it an estimator of $\theta_{t,h}^{\pi_t}$ with low error. To see this, observe that

$$\mathbb{E}_t\left[ \left((1-Y_t) + Y_t H \mathbb{1}[h=h_t^*]\right) \phi(x_{t,h}, a_{t,h}) L_{t,h} \right]$$

$$= \delta_e \mathbb{E}_t\left[ H\mathbb{1}[h=h_t^*]\phi(x_{t,h},a_{t,h})L_{t,h} \mid Y_t = 1 \right] + (1-\delta_e)\mathbb{E}_t\left[ \phi(x_{t,h},a_{t,h})L_{t,h} \mid Y_t = 0 \right]$$

$$= \delta_e \mathbb{E}_t\left[ H\mathbb{1}[h=h_t^*]\phi(x_{t,h},a_{t,h})\phi(x_{t,h},a_{t,h})^\top \theta_{t,h}^{\pi_t} \mid Y_t = 1 \right]$$

$$\qquad + (1-\delta_e)\mathbb{E}_t\left[ \phi(x_{t,h},a_{t,h})\phi(x_{t,h},a_{t,h})^\top \theta_{t,h}^{\pi_t} \mid Y_t = 0 \right]$$

**Algorithm 8** Policy Optimization with Dilated Bonuses (Linear-$Q$ Case with an Exploratory Policy)

---

**parameters**: $\lambda_{\min}, \beta, \eta, \epsilon, \delta_e, M = \left\lceil \frac{96 \ln(dHT) \ln^2(\frac{1}{\epsilon \delta_e \lambda_{\min}})}{\epsilon^2 \delta_e^2 \lambda_{\min}^2} \right\rceil, N = \left\lceil \frac{2}{\delta_e \lambda_{\min}} \ln \frac{1}{\epsilon \delta_e \lambda_{\min}} \right\rceil$.

**for** $t = 1, 2, \ldots, T$ **do**

    **Step 1: Interact with the environment.** Let $\pi_t$ be defined such that for each $x \in X_h$,

$$\pi_t(a|x) \propto \exp\left( -\eta \sum_{\tau=1}^{t-1} \left( \phi(x,a)^\top \widehat{\theta}_{\tau,h} - \text{BONUS}(\tau, x, a) \right) \right). \tag{32}$$

    Draw $Y_t \sim \text{BERNOULLI}(\delta_e)$.
    **if** $Y_t = 1$ **then**
        Draw $h_t^* \sim \text{Uniform}\{0, \ldots, H-1\}$.
        Execute $\pi_0$ in steps $0, \ldots, h_t^* - 1$; continue with $\pi_t$ in steps $h_t^*, \ldots, H-1$.
    **else** Execute $\pi_t$.

    Obtain trajectory $\{(x_{t,h}, a_{t,h}, \ell_t(x_{t,h}, a_{t,h}))\}_{h=0}^{H-1}$.

    **Step 2: Construct covariance matrix inverse estimators.**

$$\left\{ \widehat{\Sigma}_{t,h}^+ \right\}_{h=0}^{H-1} = \text{GEOMETRICRESAMPLING-MIXTURE}\,(t, M, N). \qquad \text{(see Algorithm 9)}$$

    **Step 3: Construct $Q$-function weight estimators.** For all $h = 0, \ldots, H-1$,

$$\widehat{\theta}_{t,h} = \widehat{\Sigma}_{t,h}^+ \left( (1-Y_t) + Y_t H \mathbb{1}[h=h_t^*] \right) \phi(x_{t,h}, a_{t,h}) L_{t,h}, \quad \text{where } L_{t,h} = \sum_{i=h}^{H-1} \ell_t(x_{t,i}, a_{t,i}).$$

---

$$
\begin{aligned}
&= \delta_e \mathbb{E}_t \left[ H \mathbb{1}[h=h_t^*] \right] \Sigma_h^{\pi_0} \theta_{t,h}^{\pi_t} + (1-\delta_e) \mathbb{E}_t \left[ \Sigma_{t,h} \theta_{t,h}^{\pi_t} \right] \\
&= \left( \delta_e \Sigma_h^{\pi_0} + (1-\delta_e) \Sigma_{t,h} \right) \theta_{t,h}^{\pi_t} \\
&= \Sigma_{t,h}^{\text{mix}} \theta_{t,h}^{\pi_t}
\end{aligned}
\tag{31}
$$

and thus

$$\mathbb{E}_t \left[ \widehat{\theta}_{t,h} \right] = \mathbb{E}_t \left[ \widehat{\Sigma}_{t,h}^+ \left( (1-Y_t) + Y_t H \mathbb{1}[h=h_t^*] \right) \phi(x_{t,h}, a_{t,h}) L_{t,h} \right] = \mathbb{E}_t \left[ \widehat{\Sigma}_{t,h}^+ \right] \Sigma_{t,h}^{\text{mix}} \theta_{t,h}^{\pi_t} \approx \theta_{t,h}^{\pi_t},$$

where the last step is because $\widehat{\Sigma}_{t,h}^+$ is approximately the inverse of $\Sigma_{t,h}^{\text{mix}}$.

In Appendix E.1, we first present the algorithm GEOMETRICRESAMPLING-MIXTURE and its analysis (similar to Lemma D.1). Then in Appendix E.2, we perform regret analysis for Algorithm 8.

## E.1   GEOMETRICRESAMPLING-MIXTURE

**Lemma E.1.** *Let* $M = \left\lceil \frac{96 \ln(dHT) \ln^2(\frac{1}{\epsilon \delta_e \lambda})}{\epsilon^2 \delta_e^2 \lambda^2} \right\rceil$, $N = \left\lceil \frac{2}{\delta_e \lambda} \ln \frac{1}{\epsilon \delta_e \lambda} \right\rceil$ *for some* $\epsilon > 0$. *Let* $\Sigma_{t,h} = \mathbb{E}_{\pi_t}[\phi(x_h, a_h)\phi(x_h, a_h)^\top]$ *and* $\Sigma_h^{\pi_0} = \mathbb{E}_{\pi_0}[\phi(x_h, a_h)\phi(x_h, a_h)^\top]$ *and* $\Sigma_{t,h}^{\text{mix}} = (1-\delta_e)\Sigma_{t,h} + \delta_e \Sigma_h^{\pi_0}$. *Suppose that* $\lambda > 0$ *is a lower bound for the minimum eigenvalue of* $\Sigma_h^{\pi_0}$. *Then* GEOMETRICRESAMPLING-MIXTURE *(Algorithm 9) with input* $(t, M, N)$ *ensures the following for all* $h$:

$$\left\| \widehat{\Sigma}_{t,h}^+ \right\|_{\text{op}} \le \frac{2}{\delta_e \lambda} \ln \frac{1}{\epsilon \delta_e \lambda}. \tag{33}$$

$$\left\| \mathbb{E}_t \left[ \widehat{\Sigma}_{t,h}^+ \right] - (\Sigma_{t,h}^{\text{mix}})^{-1} \right\|_{\text{op}} \le \epsilon, \tag{34}$$

$$\left\| \widehat{\Sigma}_{t,h}^+ - (\Sigma_{t,h}^{\text{mix}})^{-1} \right\|_{\text{op}} \le 2\epsilon. \tag{35}$$

$$\left\| \widehat{\Sigma}_{t,h}^+ \Sigma_{t,h}^{\text{mix}} \right\|_{\text{op}} \le 1 + 2\epsilon, \tag{36}$$

**Algorithm 9** GEOMETRICRESAMPLING-MIXTURE$(t, M, N)$

---

Let $c = \frac{1}{2}$.
**for** $m = 1, \ldots, M$ **do**
    **for** $n = 1, \ldots, N$ **do**
        With probability $1 - \delta_e$, generate path $(x_{n,0}, a_{n,0}), \ldots, (x_{n,H-1}, a_{n,H-1})$ using $\pi_t$; otherwise,
        generate it using $\pi_0$.
        For all $h$, compute $Y_{n,h} = \phi(x_{n,h}, a_{n,h})\phi(x_{n,h}, a_{n,h})^\top$.
        For all $h$, compute $Z_{n,h} = \Pi_{j=1}^n (I - cY_{j,h})$.
    For all $h$, set $\widehat{\Sigma}_{t,h}^{+(m)} = cI + c\sum_{n=1}^N Z_{n,h}$.
For all $h$, set $\widehat{\Sigma}_{t,h}^+ = \frac{1}{M}\sum_{m=1}^M \widehat{\Sigma}_{t,h}^{+(m)}$.
**return** $\widehat{\Sigma}_{t,h}^+$ for all $h = 0, \ldots, H - 1$.

---

*where $\|\cdot\|_{\mathrm{op}}$ represents the spectral norm and the last two properties Eq. (35) and Eq. (36) hold with probability at least $1 - \frac{1}{T^3}$.*

*Proof.* To prove Eq. (33), notice that each one of $\widehat{\Sigma}_{t,h}^{+(m)}$, $m = 1, \ldots, M$, is a sum of $N + 1$ terms. Furthermore, the $n$-th term of them ($cZ_{n,h}$ in Algorithm 9) has an operator norm upper bounded by $c$. Therefore,

$$\left\|\widehat{\Sigma}_{t,h}^{+(m)}\right\|_{\mathrm{op}} \le c(N+1) = \frac{1}{2}(N+1) \le \frac{2}{\delta_e\lambda}\ln\frac{1}{\epsilon\delta_e\lambda}.$$

Since $\widehat{\Sigma}_{t,h}^+$ is an average of $\widehat{\Sigma}_{t,h}^{+(m)}$, this implies Eq. (33).

To show Eq. (34), observe that

$$\mathbb{E}_t\left[\widehat{\Sigma}_{t,h}^+\right] = \mathbb{E}_t\left[\widehat{\Sigma}_{t,h}^{+(m)}\right] = cI + c\sum_{i=1}^N \left(I - c\Sigma_{t,h}^{\mathrm{mix}}\right)^i$$

$$= \left(\Sigma_{t,h}^{\mathrm{mix}}\right)^{-1}\left(I - \left(I - c\Sigma_{t,h}^{\mathrm{mix}}\right)^{N+1}\right)$$

where we use the formula: $\left(I + \sum_{i=1}^N A^i\right) = (I - A)^{-1}(I - A^{N+1})$ with $A = I - c\Sigma_{t,h}^{\mathrm{mix}}$.

Thus,

$$\left\|\mathbb{E}_t\left[\widehat{\Sigma}_{t,h}^+\right] - \left(\Sigma_{t,h}^{\mathrm{mix}}\right)^{-1}\right\|_{\mathrm{op}} = \left\|\left(\Sigma_{t,h}^{\mathrm{mix}}\right)^{-1}\left(I - c\Sigma_{t,h}^{\mathrm{mix}}\right)^{N+1}\right\|_{\mathrm{op}}$$

$$\le \frac{(1 - c\delta_e\lambda)^{N+1}}{\delta_e\lambda} \le \frac{e^{-(N+1)c\delta_e\lambda}}{\delta_e\lambda} \le \epsilon,$$

where the last inequality is by our choice of $N$ and that $c = \frac{1}{2}$.

To show Eq. (35), we only further need

$$\left\|\widehat{\Sigma}_{t,h}^+ - \mathbb{E}_t\left[\widehat{\Sigma}_{t,h}^+\right]\right\|_{\mathrm{op}} \le \epsilon$$

and combine it with Eq. (34). This can be shown by applying Lemma A.3 with $X_k = \widehat{\Sigma}_{t,h}^{+(k)}$, $\sigma = \frac{2}{\delta_e\lambda}\ln\frac{1}{\epsilon\delta_e\lambda}$, $\tau = \epsilon$, and $n = M$ (see the proof for Eq. (24) for the reason). This gives the following statement: the event $\left\|\widehat{\Sigma}_{t,h}^+ - \mathbb{E}_t\left[\widehat{\Sigma}_{t,h}^+\right]\right\|_{\mathrm{op}} > \epsilon$ holds with probability less than

$$d\exp\left(-M \times \epsilon^2 \times \frac{1}{8} \times \frac{\delta_e^2\lambda^2}{4\ln^2\frac{1}{\epsilon\delta_e\lambda}}\right) \le \frac{1}{d^2H^3T^3} \le \frac{1}{HT^3}$$

by our choice of $M$. The conclusion follows by a union bound over $h$.

To prove Eq. (36), observe that with Eq. (35), we have

$$\left\|\widehat{\Sigma}_{t,h}^+\Sigma_{t,h}^{\mathrm{mix}}\right\|_{\mathrm{op}} \le \left\|(\Sigma_{t,h}^{\mathrm{mix}})^{-1}\Sigma_{t,h}^{\mathrm{mix}}\right\|_{\mathrm{op}} + \left\|\left(\widehat{\Sigma}_{t,h}^+ - (\Sigma_{t,h}^{\mathrm{mix}})^{-1}\right)\Sigma_{t,h}^{\mathrm{mix}}\right\|_{\mathrm{op}} \le 1 + 2\epsilon$$

since $\left\|\Sigma_{t,h}^{\mathrm{mix}}\right\|_{\mathrm{op}} \le 1$. $\qquad\square$

## E.2 Regret Analysis

The analysis follows the same outline discussed in Section D.2. In particular, we define $B_t(x, a)$ for all $t, x, a$ again using the same virtual process, and then we follow the same regret decomposition as in Eq. (17), with $\widehat{Q}_t(x, a) \triangleq \phi(x, a)^\top \widehat{\theta}_{t,h}$ (for $x \in X_h$). We then bound $\mathbb{E}[\text{BIAS-1} + \text{BIAS-2}]$ and $\mathbb{E}[\text{REG-TERM}]$ in Lemma E.2 and Lemma E.3 respectively.

**Lemma E.2.** $\mathbb{E}[\text{BIAS-1} + \text{BIAS-2}] = \mathcal{O}(\epsilon H^3 T)$.

*Proof.* Consider a specific $(t, x, a)$. Let $h$ be such that $x \in X_h$. Then we have

$$
\mathbb{E}_t \left[ Q_t^{\pi_t}(x, a) - \widehat{Q}_t(x, a) \right]
$$
$$
= \phi(x, a)^\top \left( \theta_{t,h}^{\pi_t} - \mathbb{E}_t \left[ \widehat{\theta}_{t,h} \right] \right)
$$
$$
= \phi(x, a)^\top \left( \theta_{t,h}^{\pi_t} - \mathbb{E}_t \left[ \widehat{\Sigma}_{t,h}^+ \right] \mathbb{E}_t \left[ \left( (1 - Y_t) + Y_t \mathbb{1}[h_t^* = h] H \right) \phi(x_{t,h}, a_{t,h}) L_{t,h} \right] \right) + \mathcal{O}(\epsilon H^2)
$$
$$
= \phi(x, a)^\top \left( \theta_{t,h}^{\pi_t} - (\Sigma_{t,h}^{\text{mix}})^{-1} \mathbb{E}_t \left[ \left( (1 - Y_t) + Y_t \mathbb{1}[h_t^* = h] H \right) \phi(x_{t,h}, a_{t,h}) L_{t,h} \right] \right) + \mathcal{O}(\epsilon H^2)
$$
$$
\qquad \text{(by Lemma E.1 and that } \|\phi(x, a)\| \leq 1 \text{ for all } x, a \text{ and } L_{t,h} \leq H)
$$
$$
= \phi(x, a)^\top \left( \theta_{t,h}^{\pi_t} - (\Sigma_{t,h}^{\text{mix}})^{-1} \Sigma_{t,h}^{\text{mix}} \theta_{t,h}^{\pi_t} \right) + \mathcal{O}(\epsilon H^2) \qquad\qquad \text{(Eq. (31))}
$$
$$
= \mathcal{O}\left( \epsilon H^2 \right).
$$

Similarly, one can show $\mathbb{E}_t \left[ \widehat{Q}_t(x, a) - Q_t^{\pi_t}(x, a) \right] = \mathcal{O}\left( \epsilon H^2 \right)$. Summing them up over $t, x, a$ with weights $q^\star(x)\pi^\star(a|x)$ and $q^\star(x)\pi_t(a|x)$ respectively finishes the proof. $\qquad\square$

**Lemma E.3.** *If* $\eta\beta \leq \frac{\delta_e \lambda_{\min}}{24H^2 \ln(\frac{1}{\epsilon \delta_e \lambda_{\min}})}$ *and* $\eta \leq \frac{\delta_e \lambda_{\min}}{4H^2 \ln(\frac{1}{\epsilon \delta_e \lambda_{\min}})}$, *then* $\mathbb{E}[\text{REG-TERM}]$ *is upper bounded by*

$$
\frac{H \ln |A|}{\eta} + 2\eta H^3 \mathbb{E} \left[ \sum_{t=1}^T \sum_{h=0}^{H-1} \sum_{(x,a) \in X_h \times A} q^\star(x) \pi_t(a|x) \|\phi(x, a)\|_{\widehat{\Sigma}_{t,h}^+}^2 \right]
$$
$$
+ \frac{1}{H} \mathbb{E} \left[ \sum_{t=1}^T \sum_{x,a} q^\star(x) \pi_t(a|x) B_t(x, a) \right] + \widetilde{\mathcal{O}} \left( \eta \epsilon H^4 T + \frac{\eta H^4}{\delta_e^2 \lambda_{\min}^2 T^2} \right).
$$

*Proof.* The proof is similar to that of Lemma D.3. Again, we will apply the regret bound of the exponential weight algorithm Lemma A.4 for each state. We start by checking the required condition: $\eta|\phi(x, a)^\top \widehat{\theta}_{\tau,h} - B_t(x, a)| \leq 1$. This can be seen by

$$
\eta \left| \phi(x, a)^\top \widehat{\theta}_{\tau,h} \right| = \eta \left| \phi(x, a)^\top \widehat{\Sigma}_{t,h}^+ \phi(x_{t,h}, a_{t,h}) L_{t,h} \right| \times ((1 - Y_t) + Y_t \mathbb{1}[h = h^*] H)
$$
$$
\leq \eta \times \left\| \widehat{\Sigma}_{t,h}^+ \right\|_{\text{op}} \times L_{t,h} \times H
$$
$$
\leq \eta \times \frac{2}{\delta_e \lambda_{\min}} \ln \frac{1}{\epsilon \delta_e \lambda_{\min}} \times H^2 \qquad\qquad \text{(by Lemma E.1)}
$$
$$
\leq \frac{1}{2}, \qquad\qquad \text{(condition of the lemma)}
$$

and that by the definition of $\text{BONUS}(t, x, a)$, we have

$$
\eta B_t(x, a) \leq \eta \times H \left( 1 + \frac{1}{H} \right)^H \times 2\beta \sup_{x,a,h} \|\phi(x, a)\|_{\widehat{\Sigma}_{t,h}^+}^2
$$
$$
\leq 6\eta\beta \times \frac{2H}{\delta_e \lambda_{\min}} \ln \frac{1}{\epsilon \delta_e \lambda_{\min}} \qquad\qquad \text{(by Lemma E.1 again)}
$$
$$
\leq \frac{1}{2H}, \qquad\qquad (37)
$$

where the last inequality is by the first condition of the lemma.

Thus, by Lemma A.4, we have for any $x$,

$$\mathbb{E}\left[\sum_{t=1}^{T}\sum_{a}\left(\pi_t(a|x) - \pi^\star(a|x)\right)\widehat{Q}_t(x,a)\right]$$

$$\leq \frac{\ln|A|}{\eta} + 2\eta\mathbb{E}\left[\sum_{t=1}^{T}\sum_{a}\pi_t(a|x)\widehat{Q}_t(x,a)^2\right] + 2\eta\mathbb{E}\left[\sum_{t=1}^{T}\sum_{a}\pi_t(a|x)B_t(x,a)^2\right]. \quad (38)$$

The last term in Eq. (38) can be upper bounded by $\mathbb{E}\left[\frac{1}{H}\sum_{t=1}^{T}\sum_{a}\pi_t(a|x)B_t(x,a)\right]$ because $\eta B_t(x,a) \leq \frac{1}{2H}$ as we verified in Eq. (37). To bound the second term in Eq. (38), we use the following: for $(x,a) \in X_h \times A$,

$$\mathbb{E}_t\left[\widehat{Q}_t(x,a)^2\right]$$

$$\leq H^2\mathbb{E}_t\left[\phi(x,a)^\top\widehat{\Sigma}_{t,h}^+\left(((1-Y_t) + Y_tH\mathbb{1}[h=h_t^*])^2\phi(x_{t,h},a_{t,h})\phi(x_{t,h},a_{t,h})^\top\right)\widehat{\Sigma}_{t,h}^+\phi(x,a)\right]$$

$$= H^2\mathbb{E}_t\left[\phi(x,a)^\top\widehat{\Sigma}_{t,h}^+\left((1-\delta_e)\Sigma_{t,h} + \delta_e H\Sigma_h^{\pi_0}\right)\widehat{\Sigma}_{t,h}^+\phi(x,a)\right]$$

$$\leq H^3\mathbb{E}_t\left[\phi(x,a)^\top\widehat{\Sigma}_{t,h}^+\Sigma_{t,h}^{\text{mix}}\widehat{\Sigma}_{t,h}^+\phi(x,a)\right]$$

$$\leq H^3\mathbb{E}_t\left[\phi(x,a)^\top\widehat{\Sigma}_{t,h}^+\Sigma_{t,h}^{\text{mix}}(\Sigma_{t,h}^{\text{mix}})^{-1}\phi(x,a)\right] + \widetilde{\mathcal{O}}\left(\epsilon H^3 + \frac{H^3}{\delta_e^2\lambda_{\min}^2 T^3}\right) \quad (*)$$

$$= H^3\mathbb{E}_t\left[\|\phi(x,a)\|_{\widehat{\Sigma}_{t,h}^+}^2\right] + \widetilde{\mathcal{O}}\left(\epsilon H^3 + \frac{H^3}{\delta_e^2\lambda_{\min}^2 T^3}\right), \quad (39)$$

where $(*)$ is because by Eq. (35) and Eq. (36), $\left\|\widehat{\Sigma}_{t,h}^+ - (\Sigma_{t,h}^{\text{mix}})^{-1}\right\|_{\text{op}} \leq 2\epsilon$ and $\left\|\widehat{\Sigma}_{t,h}^+\Sigma_{t,h}^{\text{mix}}\right\|_{\text{op}} \leq 1 + 2\epsilon$ hold with probability $1 - \frac{1}{T^3}$; for the remaining probability, we upper bound $H^3\phi(x,a)^\top\widehat{\Sigma}_{t,h}^+\Sigma_{t,h}^{\text{mix}}\widehat{\Sigma}_{t,h}^+\phi(x,a)$ by $\frac{4H^3}{\delta_e^2\lambda_{\min}^2}\ln^2\left(\frac{1}{\epsilon\delta_e\lambda_{\min}}\right)$ using Eq. (33).

Combining them with Eq. (38) and summing over states with weights $q^\star(x)$ finishes the proof. $\square$

Finally, we are ready to prove the regret bound.

*Proof of Theorem 6.1.* Combining Lemma E.2 and Lemma E.3, we see that if we choose $\beta = 2\eta H^3$, then

$$\mathbb{E}[\text{BIAS-1} + \text{BIAS-2} + \text{REG-TERM}]$$

$$\leq \widetilde{\mathcal{O}}\left(\frac{H}{\eta} + \epsilon H^3 T + \eta\epsilon H^4 T + \frac{\eta H^4}{\delta_e^2\lambda_{\min}^2 T^2}\right) + \mathbb{E}\left[\sum_{t=1}^{T}\sum_{h=0}^{H-1}\sum_{(x,a)\in X_h\times A}q^\star(x)\pi_t(a|x)b_t(x,a)\right]$$

$$+ \frac{1}{H}\mathbb{E}\left[\sum_{t=1}^{T}\sum_{x,a}q^\star(x)\pi_t(a|x)B_t(x,a)\right].$$

Hence, by Lemma B.2, we obtain the following bound, where we first set $\epsilon = \frac{1}{H^4T}$ so that all $\epsilon$-related terms are $\widetilde{\mathcal{O}}(1)$:

$$\mathbb{E}\left[\sum_{t=1}^{T}V_t^{\pi_t}(x_0)\right] - \sum_{t=1}^{T}V_t^{\pi^\star}(x_0)$$

$$\leq \widetilde{\mathcal{O}}\left(\frac{H}{\eta} + \frac{\eta H^4}{\delta_e^2\lambda_{\min}^2 T^2} + \mathbb{E}\left[\sum_{t=1}^{T}\sum_{x,a}q_t(x,a)b_t(x,a)\right]\right)$$

$$\leq \widetilde{\mathcal{O}}\left(\frac{H}{\eta} + \frac{\eta H^4}{\delta_e^2\lambda_{\min}^2 T^2} + \beta\mathbb{E}\left[\sum_{t=1}^{T}\sum_{x,a}q_t(x,a)\|\phi(x,a)\|_{\widehat{\Sigma}_{t,h}^+}^2\right]\right)$$

$$\leq \widetilde{\mathcal{O}}\left(\frac{H}{\eta} + \frac{\eta H^4}{\delta_e^2 \lambda_{\min}^2 T^2} + \beta \mathbb{E}\left[\sum_{t=1}^{T}\sum_{x,a} q_t(x,a)\|\phi(x,a)\|_{(\Sigma_{t,h}^{\mathrm{mix}})^{-1}}^2\right]\right)$$

$$\leq \widetilde{\mathcal{O}}\left(\frac{H}{\eta} + \frac{\eta H^4}{\delta_e^2 \lambda_{\min}^2 T^2} + \frac{\beta}{1-\delta_e}\mathbb{E}\left[\sum_{t=1}^{T}\sum_{x,a} q_t(x,a)\|\phi(x,a)\|_{\Sigma_{t,h}^{-1}}^2\right]\right)$$

$$\leq \widetilde{\mathcal{O}}\left(\frac{H}{\eta} + \frac{\eta H^4}{\delta_e^2 \lambda_{\min}^2 T^2} + \beta d H T\right) \qquad\qquad \text{(Eq. (30))}$$

$$= \widetilde{\mathcal{O}}\left(\frac{H}{\eta} + \frac{\eta H^4}{\delta_e^2 \lambda_{\min}^2 T^2} + \eta d H^4 T\right). \qquad\qquad (\beta = 2\eta H^3)$$

$$(40)$$

Since we explore with probability $\delta_e$, the final regret is

$$\mathbb{E}[\mathrm{Reg}] = \widetilde{\mathcal{O}}\left(\frac{H}{\eta} + \frac{\eta H^4}{\delta_e^2 \lambda_{\min}^2 T^2} + \eta d H^4 T + \delta_e H T\right).$$

Considering the constraints specified in Lemma E.3, we choose the parameters as follows:

$$\eta = \min\left\{\frac{\delta_e \lambda_{\min}}{4H^2 \ln(\frac{1}{\epsilon\delta_e\lambda_{\min}})}, \sqrt{\frac{\delta_e \lambda_{\min}}{48H^5 \ln(\frac{1}{\epsilon\delta_e\lambda_{\min}})}}\right\},$$

$$\delta_e = \min\left\{\sqrt{\frac{H^2}{\lambda_{\min}T(\lambda_{\min}dH+1)}}, \frac{1}{2}\right\},$$

$$\epsilon = \frac{1}{H^4 T}.$$

Then the regret can be bounded by $\mathcal{O}\left(\sqrt{H^5 dT} + H^2\sqrt{\frac{T}{\lambda_{\min}}}\right)$. $\qquad\qquad\square$

## F  Details for Linear MDP with an Exploratory Policy

The algorithm for linear MDP with an exploratory policy $\pi_0$ is presented in Algorithm 10, which is based on the similar idea as Algorithm 8. Instead of changing policies on every episode, the algorithm proceeds in *epochs*, each of which consists of $W$ consecutive episodes, and the algorithm only updates its policy between epochs. We index epoch with $k$. The definitions of $\pi_k$, $\widehat{\Sigma}_{k,h}^+$, $B_k(x,a)$ are analogous to those of $\pi_t$, $\widehat{\Sigma}_{t,h}^+$, $B_t(x,a)$ in previous sections.

To deal with the epoch-based update, we define the following quantities (notice that the $k$-th epoch consists of episodes $(k-1)W+1,\ldots,kW$):

**Definition 1.**

$$\overline{\ell}_k(x,a) \triangleq \frac{1}{W}\sum_{t=(k-1)W+1}^{kW} \ell_t(x,a)$$

$$\overline{Q}_k^{\pi}(x,a) \triangleq Q^{\pi}(x,a;\overline{\ell}_k)$$

$$\overline{\theta}_{k,h}^{\pi} \triangleq \frac{1}{W}\sum_{t=(k-1)W+1}^{kW} \theta_{t,h}^{\pi}$$

Recall that the main difference between Algorithm 10 and Algorithm 8 is that in Algorithm 10 we use linear function approximation to calculate the bonus. The bonus $B_k(x,a)$ and the *estimated* bonus $\widehat{\Lambda}_k(x,a)$ are defined in Definition 2.

**Definition 2.**

$$B_k(x,a) \triangleq b_k(x,a) + \left(1+\frac{1}{H}\right)\mathbb{E}_{x'\sim P(\cdot|x,a)}\mathbb{E}_{a'\sim\pi_t(\cdot|x')}[B_t(x',a')] \quad \text{(with } B_k(x_H,a) \triangleq 0)$$

$$\widehat{B}_k(x,a) \triangleq b_k(x,a) + \phi(x,a)^{\top}\widehat{\Lambda}_{k,h} \qquad \text{(for } x \in X_h)$$

where $b_k(x,a)$ and $\widehat{\Lambda}_{k,h}$ are defined in Algorithm 10.

**Algorithm 10** Policy Optimization with Dilated Bonuses (Linear MDP with an Exploratory Policy)

**Parameters**: $\lambda_{\min}, \beta, \eta, \epsilon, M = \left\lceil \frac{96 \ln(dHT) \ln^2(\frac{1}{\epsilon \delta_e \lambda_{\min}})}{\epsilon^2 \delta_e^2 \lambda_{\min}^2} \right\rceil, N = \left\lceil \frac{2}{\delta_e \lambda_{\min}} \ln \frac{1}{\epsilon \delta_e \lambda_{\min}} \right\rceil, W = 2MN$

**for** $k = 1, 2, \ldots, T/W$ **do**

    **1) Interact with the environment**: Define $\pi_k$ as the following for $x \in X_h$:

$$\pi_k(a|x) \propto \exp\left(-\eta \sum_{\tau=1}^{k-1} \left(\phi(x,a)^\top \widehat{\theta}_{\tau,h} - \phi(x,a)^\top \widehat{\Lambda}_{\tau,h} - b_\tau(x,a)\right)\right) \tag{41}$$

    where

$$b_\tau(x,a) \triangleq \beta \|\phi(x,a)\|_{\widehat{\Sigma}_{\tau,h}^+}^2 + \beta \sum_{a'} \pi_\tau(a'|x) \|\phi(x,a')\|_{\widehat{\Sigma}_{\tau,h}^+}^2 .$$

    Randomly divide $[(k-1)W + 1, \ldots, kW]$ into two parts: $S$ and $S'$, such that $|S| = |S'| = W/2$.

    **for** $t = (k-1)W + 1, \ldots, kW$ **do**
        Draw $Y_t \sim \text{BERNOULLI}(\delta_e)$.
        **if** $Y_t = 1$ *and* $t \in S$ **then** Execute $\pi_0$
        **else if** $Y_t = 1$ *and* $t \in S'$ **then**
            Draw $h_t^* \sim \text{Uniform}\{0, \ldots, H-1\}$.
            Execute $\pi_0$ in steps $0, \ldots, h_t^* - 1$; continue with $\pi_t$ in steps $h_t^*, \ldots, H-1$.
        **else** Execute $\pi_t$
        Collect trajectories $\{(x_{t,h}, a_{t,h}, \ell_t(x_{t,h}, a_{t,h}))\}_{h=0}^{H-1}$

    **2) Construct inverse covariance matrix estimators:** Use the samples in $S$ to calculate the following (note that $|S| = W/2 = MN$ and the GEOMETRICRESAMPLING-MIXTURE requires exactly $MN$ episodes of samples. We simply view these $MN$ episodes as calls within GEOMETRICRESAMPLING-MIXTURE):

$$\left\{\widehat{\Sigma}_{k,h}^+\right\}_{h=0}^{H-1} = \text{GEOMETRICRESAMPLING-MIXTURE}(k, M, N). \tag{42}$$

    **3) Construct Q-function estimators:** Define for all $t, h$:

$$L_{t,h} \triangleq \sum_{i=h}^{H-1} \ell_t(x_{t,i}, a_{t,i})$$

    and

$$\widehat{\theta}_{k,h} \triangleq \widehat{\Sigma}_{k,h}^+ \left(\frac{1}{|S'|} \sum_{t \in S'} ((1 - Y_t) + Y_t H \mathbb{1}[h = h_t^*]) \phi(x_{t,h}, a_{t,h}) L_{t,h}\right). \tag{43}$$

    **4) Construct bonus function estimators:** Define for all $t, h$:

$$D_{t,h} \triangleq \begin{cases} 0 & \text{if } h = H-1, \\ \sum_{i=h+1}^{H-1} \left(1 + \frac{1}{H}\right)^{i-h} b_t(x_{t,i}, a_{t,i}) & \text{otherwise;} \end{cases}$$

    and

$$\widehat{\Lambda}_{k,h} \triangleq \widehat{\Sigma}_{k,h}^+ \left(\frac{1}{|S'|} \sum_{t \in S'} ((1 - Y_t) + Y_t H \mathbb{1}[h = h_t^*]) \phi(x_{t,h}, a_{t,h}) D_{t,h}\right). \tag{44}$$

## F.1 Regret Analysis

The regret decomposition for this section is slightly different from those in previous sections. Since we also use function approximation on the bonus $B_t(x, a)$, we need to also account for its estimation error, resulting in two extra bias terms:

$$\sum_{k=1}^{T/W} \sum_x q^\star(x) \left\langle \pi_k(\cdot|x) - \pi^\star(\cdot|x), \overline{Q}_k^{\pi_k}(x, \cdot) - B_k(x, \cdot) \right\rangle$$

$$= \underbrace{\sum_{k=1}^{T/W} \sum_x q^\star(x) \left\langle \pi_k(\cdot|x), \overline{Q}_k^{\pi_k}(x, \cdot) - \widehat{Q}_k(x, \cdot) \right\rangle}_{\text{BIAS-1}} + \underbrace{\sum_{k=1}^{T/W} \sum_x q^\star(x) \left\langle \pi^\star(\cdot|x), \widehat{Q}_k(x, \cdot) - \overline{Q}_k^{\pi_k}(x, \cdot) \right\rangle}_{\text{BIAS-2}}$$

$$+ \underbrace{\sum_{k=1}^{T/W} \sum_x q^\star(x) \left\langle \pi_k(\cdot|x), \widehat{B}_k(x, \cdot) - B_k(x, \cdot) \right\rangle}_{\text{BIAS-3}} + \underbrace{\sum_{k=1}^{T/W} \sum_x q^\star(x) \left\langle \pi^\star(\cdot|x), B_k(x, \cdot) - \widehat{B}_k(x, \cdot) \right\rangle}_{\text{BIAS-4}}$$

$$+ \underbrace{\sum_{k=1}^{T/W} \sum_x q^\star(x) \left\langle \pi_k(\cdot|x) - \pi^\star(\cdot|x), \widehat{Q}_k(x, \cdot) - \widehat{B}_k(x, \cdot) \right\rangle}_{\text{REG-TERM}}$$

In the following lemmas, we bound each term separately:

**Lemma F.1.**

$$\mathbb{E}[\text{BIAS-1} + \text{BIAS-2}] \leq \mathcal{O}\left(\frac{\epsilon H^3 T}{W}\right).$$

*Proof.* The proof of this lemma is almost identical to that of Lemma E.2, except that we replace $T$ by $T/W$, and consider the averaged loss $\overline{\ell}_k$ in an epoch instead of the single episode loss $\ell_t$:

$$\mathbb{E}_k \left[ \overline{Q}_k^{\pi_k}(x, a) - \widehat{Q}_k(x, a) \right]$$

$$= \phi(x, a)^\top \left( \overline{\theta}_{k,h}^{\pi_k} - \mathbb{E}_k \left[ \widehat{\theta}_{k,h} \right] \right)$$

$$= \phi(x, a)^\top \left( \overline{\theta}_{k,h}^{\pi_k} - \mathbb{E}_k \left[ \widehat{\Sigma}_{k,h}^+ \right] \mathbb{E}_k \left[ \frac{1}{|S'_k|} \sum_{t \in S'_k} ((1 - Y_t) + Y_t H \mathbb{1}[h = h_t^*]) \phi(x_{t,h}, a_{t,h}) L_{t,h} \right] \right)$$

$$(S'_k \text{ is the } S' \text{ in Algorithm 10 within epoch } k)$$

$$= \phi(x, a)^\top \left( \overline{\theta}_{k,h}^{\pi_k} - (\Sigma_{k,h}^{\text{mix}})^{-1} \mathbb{E}_k \left[ \frac{1}{|S'_k|} \sum_{t \in S'_k} ((1 - Y_t) + Y_t H \mathbb{1}[h = h_t^*]) \phi(x_{t,h}, a_{t,h}) L_{t,h} \right] \right) + \mathcal{O}(\epsilon H^2)$$

$$(\text{by Lemma E.1 and that } \|\phi(x, a)\| \leq 1 \text{ for all } x, a \text{ and } L_{t,h} \leq H)$$

$$= \phi(x, a)^\top \left( \overline{\theta}_{k,h}^{\pi_k} - (\Sigma_{k,h}^{\text{mix}})^{-1} \mathbb{E}_k \left[ \frac{1}{|S'_k|} \sum_{t \in S'_k} \Sigma_{k,h}^{\text{mix}} \theta_{t,h}^{\pi_k} \right] \right) + \mathcal{O}(\epsilon H^2) \tag{45}$$

$$= \phi(x, a)^\top \left( \overline{\theta}_{k,h}^{\pi_k} - (\Sigma_{k,h}^{\text{mix}})^{-1} \mathbb{E}_k \left[ \frac{1}{W} \sum_{t=(k-1)W+1}^{kW} \Sigma_{k,h}^{\text{mix}} \theta_{t,h}^{\pi_k} \right] \right) + \mathcal{O}(\epsilon H^2)$$

$$(S'_k \text{ is randomly chosen from epoch } k)$$

$$= \phi(x, a)^\top \left( \overline{\theta}_{k,h}^{\pi_k} - (\Sigma_{k,h}^{\text{mix}})^{-1} \Sigma_{k,h}^{\text{mix}} \overline{\theta}_{k,h}^{\pi_k} \right) + \mathcal{O}(\epsilon H^2)$$

$$= \mathcal{O}\left(\epsilon H^2\right) \tag{46}$$

Similarly, $\mathbb{E}_k\left[\widehat{Q}_k(x,a) - \overline{Q}_k^{\pi_k}(x,a)\right] = \mathcal{O}\left(\epsilon H^2\right)$. Summing them over $k, x, a$ using weights $q^\star(x)\pi_k(a|x)$ and $q^\star(x)\pi^\star(a|x)$ respectively finishes the proof. $\qquad\square$

**Lemma F.2.**

$$\mathbb{E}[\text{BIAS-}3 + \text{BIAS-}4] \leq \widetilde{\mathcal{O}}\left(\frac{\epsilon H^3 T}{W} \times \frac{\beta}{\delta_e \lambda_{\min}}\right).$$

*Proof.* The proof is almost identical to that of the previous lemma. Recall the definition of $\Lambda_{k,h}^{\pi_k}$ in Section 6. Then we have

$$\mathbb{E}_k\left[B_k(x,a) - \widehat{B}_k(x,a)\right]$$

$$= \phi(x,a)^\top \left(\Lambda_{k,h}^{\pi_k} - \mathbb{E}_k\left[\widehat{\Lambda}_{k,h}\right]\right)$$

$$= \phi(x,a)^\top \left(\Lambda_{k,h}^{\pi_k} - \mathbb{E}_k\left[\widehat{\Sigma}_{k,h}^+\right]\mathbb{E}_k\left[\frac{1}{|S'_k|}\sum_{t \in S'_k}((1 - Y_t) + Y_t H\mathbb{1}[h = h_t^*])\phi(x_{t,h}, a_{t,h})D_{t,h}\right]\right)$$

$$\text{($S'_k$ is the $S'$ in Algorithm 10 within epoch $k$)}$$

$$= \phi(x,a)^\top \left(\Lambda_{k,h}^{\pi_k} - \left(\Sigma_{k,h}^{\text{mix}}\right)^{-1}\mathbb{E}_k\left[\frac{1}{|S'_k|}\sum_{t \in S'_k}((1 - Y_t) + Y_t H\mathbb{1}[h = h_t^*])\phi(x_{t,h}, a_{t,h})D_{t,h}\right]\right)$$

$$+ \widetilde{\mathcal{O}}\left(\epsilon H^2 \times \frac{\beta}{\delta_e \lambda_{\min}}\right)$$

(by Lemma E.1 and that $\|\phi(x,a)\| \leq 1$ for all $x,a$ and $D_{t,h} = \mathcal{O}(H\beta \sup\|\widehat{\Sigma}_{k,h}^+\|_{\text{op}}) = \widetilde{\mathcal{O}}(\frac{H\beta}{\delta_e \lambda_{\min}})$)

$$= \phi(x,a)^\top \left(\Lambda_{k,h}^{\pi_k} - \left(\Sigma_{k,h}^{\text{mix}}\right)^{-1}\mathbb{E}_k\left[\frac{1}{|S'_k|}\sum_{t \in S'_k}\Sigma_{k,h}^{\text{mix}}\Lambda_{k,h}^{\pi_k}\right]\right) + \widetilde{\mathcal{O}}\left(\epsilon H^2 \times \frac{\beta}{\delta_e \lambda_{\min}}\right)$$

$$= \phi(x,a)^\top \left(\Lambda_{k,h}^{\pi_k} - \left(\Sigma_{k,h}^{\text{mix}}\right)^{-1}\Sigma_{k,h}^{\text{mix}}\Lambda_{k,h}^{\pi_k}\right) + \widetilde{\mathcal{O}}\left(\epsilon H^2 \times \frac{\beta}{\delta_e \lambda_{\min}}\right)$$

$$= \widetilde{\mathcal{O}}\left(\epsilon H^2 \times \frac{\beta}{\delta_e \lambda_{\min}}\right). \tag{47}$$

Similar for $\mathbb{E}_k[\widehat{B}_k(x,a) - B_k(x,a)]$. Summing them over $k, x, a$ using weights $q^\star(x)\pi^\star(a|x)$ and $q^\star(x)\pi_t(a|x)$ respectively /finishes the proof. $\qquad\square$

**Lemma F.3.** *Let* $\frac{\eta\beta}{\delta_e^2 \lambda_{\min}^2} \leq \frac{1}{160 H^4 \ln(\frac{1}{\epsilon\delta_e\lambda_{\min}})^2}$ *and* $\frac{\eta}{\delta_e\lambda_{\min}} \leq \frac{1}{4H^2 \ln(\frac{1}{\epsilon\delta_e\lambda_{\min}})}$. *Then*

$$\mathbb{E}[\text{REG-TERM}]$$

$$= \widetilde{\mathcal{O}}\left(\frac{H}{\eta} + \frac{\eta\epsilon H^4 T}{W} + \frac{\eta H^4}{\delta_e^2 \lambda_{\min}^2 T^2 W} + \frac{\eta\epsilon\beta^2 H^4 T}{\delta_e^2 \lambda_{\min}^2 W} + \frac{\eta H^4 \beta^2}{\delta_e^3 \lambda_{\min}^3 T^2 W}\right)$$

$$+ 2\eta H^3 \mathbb{E}\left[\sum_{k,x,a} q^\star(x)\pi_k(x,a)\|\phi(x,a)\|_{\widehat{\Sigma}_{k,h}^+}^2\right] + \frac{1}{H}\mathbb{E}\left[\sum_{k,x,a} q^\star(x)\pi_k(a|x)B_k(x,a)\right].$$

*Proof.* We first check the condition for Lemma A.4: $\eta\left|\widehat{Q}_k(x,a) - \widehat{B}_t(x,a)\right| \leq 1$. In our case,

$$\eta\left|\widehat{Q}_k(x,a)\right| = \eta\left|\phi(x,a)^\top \widehat{\Sigma}_{k,h}^+\left(\frac{1}{|S'|}\sum_{t \in S'}((1 - Y_t) + Y_t H\mathbb{1}[h = h_t^*])\phi(x_{t,h}, a_{t,h})L_{t,h}\right)\right|$$

$$\leq \eta \times \|\widehat{\Sigma}_{k,h}^+\|_{\text{op}} \times H \times \sup_{t \in S'} L_{t,h}$$

$$\leq \eta \times \frac{2}{\delta_e \lambda_{\min}} \ln \frac{1}{\epsilon\delta_e\lambda_{\min}} \times H^2 \qquad \text{(by Lemma E.1)}$$

$$\le \frac{1}{2} \qquad\qquad\text{(by the condition specified in the lemma)}$$

and

$$\eta\left|\widehat{B}_k(x,a)\right| \le \eta\left|b_k(x,a)\right| + \eta\left|\phi(x,a)^\top\widehat{\Sigma}_{k,h}^+\left(\frac{1}{|S'|}\sum_{t\in S'}((1-Y_t)+Y_tH\mathbb{1}[h=h_t^*])\phi(x_{t,h},a_{t,h})D_{t,h}\right)\right|$$

$$\le \eta\times 2\beta\times\|\widehat{\Sigma}_{k,h}^+\|_{\mathrm{op}} + \eta\times\|\widehat{\Sigma}_{k,h}^+\|_{\mathrm{op}}\times H\times\sup_{t\in S'}D_{t,h}$$

$$\le \eta\times 2\beta\times\|\widehat{\Sigma}_{k,h}^+\|_{\mathrm{op}} + \eta\times\|\widehat{\Sigma}_{k,h}^+\|_{\mathrm{op}}\times H\times(H-1)\left(1+\frac{1}{H}\right)^H\times 2\beta\|\widehat{\Sigma}_{k,h}^+\|_{\mathrm{op}}$$

$$\le 8\eta\beta H^2\times\|\widehat{\Sigma}_{k,h}^+\|_{\mathrm{op}}^2$$

$$\le 8\eta\beta H^2\left(\frac{2}{\delta_e\lambda_{\min}}\ln\frac{1}{\epsilon\delta_e\lambda_{\min}}\right)^2 \qquad\qquad\text{(by Lemma E.1)}$$

$$\le \frac{1}{2H}. \qquad\qquad\text{(by the condition specified in the lemma)}$$

An upper bound for $\mathbb{E}_k\left[\widehat{Q}_k(x,a)^2\right]$ follows the same calculation as in Eq. (39):

$$\mathbb{E}_k\left[\widehat{Q}_k(x,a)^2\right]$$

$$\le \mathbb{E}_k\left[\frac{1}{|S_k'|}\sum_{t\in S_k'}H^2\phi(x,a)^\top\widehat{\Sigma}_{k,h}^+\left(((1-Y_t)+Y_tH\mathbb{1}[h=h_t^*])^2\phi(x_{t,h},a_{t,h})\phi(x_{t,h},a_{t,h})^\top\right)\widehat{\Sigma}_{k,h}^+\phi(x,a)\right]$$

$$(*)$$

$$= \mathbb{E}_k\left[H^2\phi(x,a)^\top\widehat{\Sigma}_{k,h}^+\left((1-\delta_e)\Sigma_{k,h}+\delta_eH\Sigma_h^{\pi_0}\right)\widehat{\Sigma}_{k,h}^+\phi(x,a)\right]$$

$$\le H^3\mathbb{E}_k\left[\phi(x,a)^\top\widehat{\Sigma}_{k,h}^+\Sigma_{k,h}^{\mathrm{mix}}\widehat{\Sigma}_{k,h}^+\phi(x,a)\right]$$

$$\le H^3\mathbb{E}_k\left[\phi(x,a)^\top\widehat{\Sigma}_{k,h}^+\Sigma_{k,h}^{\mathrm{mix}}(\Sigma_{k,h}^{\mathrm{mix}})^{-1}\phi(x,a)\right] + \widetilde{\mathcal{O}}\left(\epsilon H^3+\frac{H^3}{\delta_e^2\lambda_{\min}^2T^3}\right)$$

$$\text{(by Eq. (35) and Eq. (36))}$$

$$= H^3\mathbb{E}_k\left[\|\phi(x,a)\|_{\widehat{\Sigma}_{k,h}^+}^2\right] + \widetilde{\mathcal{O}}\left(\epsilon H^3+\frac{H^3}{\delta_e^2\lambda_{\min}^2T^3}\right), \tag{48}$$

where in $(*)$ we use $\left(\frac{1}{|S_k'|}\sum_{t\in S_k'}v_t\right)^2 \le \frac{1}{|S_k'|}\sum_{t\in S_k'}v_t^2$ with $v_t = \phi(x,a)^\top\widehat{\Sigma}_{k,h}^+\left((1-Y_t)+Y_tH\mathbb{1}[h=h_t^*]\right)\phi(x_{t,h},a_{t,h})L_{t,h}$.

Next, we bound $\mathbb{E}_t\left[\widehat{B}_t(x,a)^2\right]$:

$$\mathbb{E}_k\left[\widehat{B}_k(x,a)^2\right]$$

$$\le 2\mathbb{E}_k\left[b_k(x,a)^2\right] + 2\mathbb{E}_k\left[(\phi(x,a)^\top\widehat{\Lambda}_{k,h})^2\right]$$

$$\le 2\left(\beta\|\phi(x,a)\|_{\widehat{\Sigma}_{k,h}^+}^2 + \beta\mathbb{E}_{a'\sim\pi_k(\cdot|x)}\left[\|\phi(x,a')\|_{\widehat{\Sigma}_{k,h}^+}^2\right]\right)^2$$

$$+ 2H^3\left(\frac{6\beta}{\delta_e\lambda_{\min}}\ln\frac{1}{\epsilon\delta_e\lambda_{\min}}\right)^2\mathbb{E}_k\left[\|\phi(x,a)\|_{\widehat{\Sigma}_{k,h}^+}^2\right] + \widetilde{\mathcal{O}}\left(\epsilon H^3+\frac{H^3}{\delta_e^2\lambda_{\min}^2T^3}\right)\times\left(\frac{6\beta}{\delta_e\lambda_{\min}}\ln\frac{1}{\epsilon\delta_e\lambda_{\min}}\right)^2$$

$$\le \frac{8\beta}{\delta_e\lambda_{\min}}\ln\frac{1}{\epsilon\delta_e\lambda_{\min}}\left(\beta\|\phi(x,a)\|_{\widehat{\Sigma}_{k,h}^+}^2 + \beta\mathbb{E}_{a'\sim\pi_k(\cdot|x)}\left[\|\phi(x,a')\|_{\widehat{\Sigma}_{k,h}^+}^2\right]\right)$$

$$+ 2H^3\left(\frac{6\beta}{\delta_e\lambda_{\min}}\ln\frac{1}{\epsilon\delta_e\lambda_{\min}}\right)^2\mathbb{E}_k\left[\|\phi(x,a)\|_{\widehat{\Sigma}_{k,h}^+}^2\right] + \widetilde{\mathcal{O}}\left(\epsilon H^3+\frac{H^3}{\delta_e^2\lambda_{\min}^2T^3}\right)\times\left(\frac{6\beta}{\delta_e\lambda_{\min}}\ln\frac{1}{\epsilon\delta_e\lambda_{\min}}\right)^2$$

$$\leq \left( \frac{8\beta}{\delta_e \lambda_{\min}} \ln \frac{1}{\epsilon \delta_e \lambda_{\min}} + \frac{72 H^3 \beta}{\delta_e^2 \lambda_{\min}^2} \ln^2 \left( \frac{1}{\epsilon \delta_e \lambda_{\min}} \right) \right) b_k(x,a) + \widetilde{\mathcal{O}} \left( \frac{\epsilon \beta^2 H^3}{\delta_e^2 \lambda_{\min}^2} + \frac{H^3 \beta^2}{\delta_e^4 \lambda_{\min}^4 T^3} \right)$$

$$\leq \frac{80 H^3 \beta}{\delta_e^2 \lambda_{\min}^2} \ln^2 \left( \frac{1}{\epsilon \delta_e \lambda_{\min}} \right) b_k(x,a) + \widetilde{\mathcal{O}} \left( \frac{\epsilon \beta^2 H^3}{\delta_e^2 \lambda_{\min}^2} + \frac{H^3 \beta^2}{\delta_e^4 \lambda_{\min}^4 T^3} \right),$$

where in the second inequality we bound $\mathbb{E}_k \left[ (\phi(x,a)^\top \widehat{\Lambda}_{k,h})^2 \right]$ similarly as we bound $\mathbb{E}_k \left[ \widehat{Q}_k(x,a)^2 \right]$ in Eq. (48), except that we replace the upper bound for $L_{t,h}$ as $H$ by the upper bound for $D_{t,h}$ as $H \left( 1 + \frac{1}{H} \right)^H \beta \|\widehat{\Sigma}_{k,h}^+\|_{\mathrm{op}} \leq 3H \times \beta \times \frac{2}{\delta_e \lambda_{\min}} \ln \frac{1}{\epsilon \delta_e \lambda_{\min}}$ by Eq. (33). In the third inequality, we use that

$$\beta \|\phi(x,a)\|_{\widehat{\Sigma}_{k,h}^+}^2 \leq \beta \times \frac{2}{\delta_e \lambda_{\min}} \ln \frac{1}{\epsilon \delta_e \lambda_{\min}}. \qquad \text{(also by Eq. (33))}$$

Thus, by Lemma A.4, we have

$\mathbb{E}[\text{REG-TERM}]$

$$\leq \widetilde{\mathcal{O}} \left( \frac{H}{\eta} \right) + 2\eta \sum_{k,x,a} q^\star(x) \pi_k(a|x) \widehat{Q}_k(x,a)^2 + 2\eta \sum_{k,x,a} q^\star(x) \pi_k(a|x) B_k(x,a)^2$$

$$\leq \widetilde{\mathcal{O}} \left( \frac{H}{\eta} + \frac{\eta \epsilon H^4 T}{W} + \frac{\eta H^4}{\delta_e^2 \lambda_{\min}^2 T^2 W} + \frac{\eta \epsilon \beta^2 H^4 T}{\delta_e^2 \lambda_{\min}^2 W} + \frac{\eta H^4 \beta^2}{\delta_e^4 \lambda_{\min}^4 T^2 W} \right)$$

$$+ 2\eta H^3 \mathbb{E} \left[ \sum_{k,x,a} q^\star(x) \pi_k(x,a) \|\phi(x,a)\|_{\widehat{\Sigma}_{k,h}^+}^2 \right] + \frac{160 H^3 \eta \beta}{\delta_e^2 \lambda_{\min}^2} \ln^2 \left( \frac{1}{\epsilon \delta_e \lambda_{\min}} \right) \mathbb{E} \left[ \sum_{k,x,a} q^\star(x) \pi_k(a|x) b_k(x,a) \right]$$

$$\leq \widetilde{\mathcal{O}} \left( \frac{H}{\eta} + \frac{\eta \epsilon H^4 T}{W} + \frac{\eta H^4}{\delta_e^2 \lambda_{\min}^2 T^2 W} + \frac{\eta \epsilon \beta^2 H^4 T}{\delta_e^2 \lambda_{\min}^2 W} + \frac{\eta H^4 \beta^2}{\delta_e^4 \lambda_{\min}^4 T^2 W} \right)$$

$$+ 2\eta H^3 \mathbb{E} \left[ \sum_{k,x,a} q^\star(x) \pi_k(x,a) \|\phi(x,a)\|_{\widehat{\Sigma}_{k,h}^+}^2 \right] + \frac{1}{H} \mathbb{E} \left[ \sum_{k,x,a} q^\star(x) \pi_k(a|x) B_k(x,a) \right]$$

where in the last inequality we use the condition specified in the lemma and that $B_k(x,a) \geq b_k(x,a)$.

$\square$

*Proof of Theorem 6.2.* Now we combine the bounds in Lemma F.2, Lemma F.2, Lemma F.3. We get

$\mathbb{E}[\text{BIAS-1} + \text{BIAS-2} + \text{BIAS-3} + \text{BIAS-4} + \text{REG-TERM}]$

$$= \widetilde{\mathcal{O}} \left( \frac{H}{\eta} + \frac{\eta \epsilon H^4 T}{W} + \frac{\eta H^4}{\delta_e^2 \lambda_{\min}^2 T^2 W} + \frac{\eta \epsilon \beta^2 H^4 T}{\delta_e^2 \lambda_{\min}^2 W} + \frac{\eta H^4 \beta^2}{\delta_e^4 \lambda_{\min}^4 T^2 W} + \frac{\epsilon H^3 T}{W} \times \frac{\beta}{\delta_e \lambda_{\min}} + \frac{\epsilon H^3 T}{W} \right)$$

$$+ \sum_{k,x,a} q^\star(x) \pi_k(a|x) b_k(x,a)$$

$$+ \frac{1}{H} \sum_{k,x,a} q^\star(x) \pi_k(a|x) B_k(x,a)$$

where we use $b_k = 2\eta H^3$. By picking $\beta \leq \delta_e \lambda_{\min}$, the first term above can be further upper bounded by

$$\widetilde{\mathcal{O}} \left( \frac{H}{\eta} + \frac{\eta \epsilon H^4 T}{W} + \frac{\eta H^4}{\delta_e^2 \lambda_{\min}^2 T^2 W} + \frac{\epsilon H^3 T}{W} \right).$$

By Lemma B.2, we have

$$\mathbb{E} \left[ \sum_{k=1}^{T/W} V^{\pi_k}(x_0; \overline{\ell}_k) \right] - \sum_{k=1}^{T/W} V^{\pi^\star}(x_0; \overline{\ell}_k)$$

$$\leq \widetilde{\mathcal{O}} \left( \frac{H}{\eta} + \frac{\eta \epsilon H^4 T}{W} + \frac{\eta H^4}{\delta_e^2 \lambda_{\min}^2 T^2 W} + \frac{\epsilon H^3 T}{W} + \beta \mathbb{E} \left[ \sum_{k=1}^{\lceil T/W \rceil} \sum_{x,a} q_k(x) \pi_k(a|x) \|\phi(x,a)\|_{\widehat{\Sigma}_{k,h}^+}^2 \right] \right)$$

$$= \widetilde{\mathcal{O}} \left( \frac{H}{\eta} + \frac{\eta \epsilon H^4 T}{W} + \frac{\eta H^4}{\delta_e^2 \lambda_{\min}^2 T^2 W} + \frac{\epsilon H^3 T}{W} + \frac{\beta d H T}{W} \right)$$

(by similar calculation as in Eq. (40))

$$= \widetilde{\mathcal{O}} \left( \frac{H}{\eta} + \frac{\eta \epsilon H^4 T}{W} + \frac{\eta H^4}{\delta_e^2 \lambda_{\min}^2 T^2 W} + \frac{\epsilon H^3 T}{W} + \frac{\eta d H^4 T}{W} \right) \qquad (\beta = 2\eta H^3)$$

Multiplying back with $W$, and considering the exploration rate $\delta_e$, we see that the true expected regret is upper bounded by

$$\widetilde{\mathcal{O}} \left( \frac{HW}{\eta} + \eta \epsilon H^4 T + \frac{\eta H^4}{\delta_e^2 \lambda_{\min}^2 T^2} + \epsilon H^3 T + \eta d H^4 T + \delta_e H T \right)$$

$$= \widetilde{\mathcal{O}} \left( \frac{H}{\eta \epsilon^2 \delta_e^3 \lambda_{\min}^3} + \eta \epsilon H^4 T + \frac{\eta H^4}{\delta_e^2 \lambda_{\min}^2 T^2} + \epsilon H^3 T + \eta d H^4 T + \delta_e H T \right)$$

where we use the specified value of $M$ and $N$ and that $W = 2MN$.

Considering the constraints in Lemma F.3 and that $\beta = 2\eta H^3$, we pick

$$\eta = \frac{\delta_e \lambda_{\min}}{20 H^{3.5} \ln \left( \frac{1}{\epsilon \delta_e \lambda_{\min}} \right)}$$

which also makes $\beta \leq \delta_e \lambda_{\min}$ as we assumed previously.

With this $\eta$, the regret can be simplified as

$$\widetilde{\mathcal{O}} \left( \frac{H^{4.5}}{\epsilon^2 \delta_e^4 \lambda_{\min}^4} + \delta_e \lambda_{\min} \epsilon \sqrt{H} T + \frac{\sqrt{H}}{\delta_e \lambda_{\min} T^2} + \epsilon H^3 T + \delta_e \lambda_{\min} d \sqrt{H} T + \delta_e H T \right)$$

$$= \widetilde{\mathcal{O}} \left( \frac{H^{4.5}}{\epsilon^2 \delta_e^4 \lambda_{\min}^4} + \epsilon H^3 T + \delta_e \lambda_{\min} d \sqrt{H} T + \delta_e H T \right)$$

By picking

$$\epsilon = \left( \frac{H^{1.5}}{\delta_e^4 \lambda_{\min}^4 T} \right)^{\frac{1}{3}}, \qquad \delta_e = \left( \frac{H^9}{T \lambda_{\min}^4 (\lambda_{\min} d + \sqrt{H})^3} \right)^{\frac{1}{7}},$$

we get a regret bound of

$$\widetilde{\mathcal{O}} \left( \left( H^{12.5} \left( \frac{d \lambda_{\min} + \sqrt{H}}{\lambda_{\min}} \right)^4 T^6 \right)^{\frac{1}{7}} \right).$$

□