# OpenReview forum: "Policy Optimization in Adversarial MDPs: Improved Exploration via Dilated Bonuses"
_NeurIPS.cc/2021/Conference — NeurIPS 2021 Poster_

### Official Review · Reviewer_GhdY · 2021-07-12

**Rating:** 6
**Confidence:** 4

**Summary:**

The paper studies the regret of policy optimization for adversarial MDPs with bandit feedback. It develops a general solution that adds dilated bonuses to the policy update for exploration and applies the algorithm to the tabular, linear-Q, and linear MDP settings.   It shows that such a solution improves and generalizes the state-of-the-art.


**Limitations And Societal Impact:**

- The paper can include a table of the comparison of this work with previous works.

- The paper does not provide any empirical study to show that the dilated bonus works better. I understand this paper focuses on the theory but some toy examples could be helpful to show the benefit of the dilated bonus.


**Main Review:**

- (+) For the tabular setting, the paper improved the regret analysis to $O(\sqrt{T})$ with dilated bonus. The idea is novel and the theoretical contribution is solid.

- (+) For the linear-Q setting, the paper establishes a regret of $O(T^{2/3})$ with the assumption on the existence of a simulator and $O(T^{1/2})$ with an exploratory policy. The assumptions are weaker than the previous work. The theoretical contribution of this part is solid.

- (+) The paper also provides regret for linear MDP without a simulator.

- (+) The mechanism of the dilated bonus is well discussed in Section 3.


- (-) The notation of $Q(x_h, a)$ and $V(x_h) $ are used without specification.  Such notations of value functions with specifying the timestep $h$ by subscription of $x$ is confusing. I suggest the authors to use e.g., $V(x, h)$ and $Q(x, a, h)$.



**Time Spent Reviewing:**

4

---

> ### Author Response · Authors · 2021-08-09
> **Response**
>
> Thank you for the constructive suggestions for our work. See our responses below:
>
> Q1: The notation of $Q(x_h, a)$ and $V(x_h)$ are used without specifying the timestep index $h$.
>
> A1:  In fact, as we assume that the state space $X$ is partitioned into disjoint sets $X_0, X_1, \ldots, X_H$ (see Line 80 in Section 2), the state $x$ itself contains the information of its corresponding layer index already, and thus $Q(x,a)$ and $V(x)$ can be used without ambiguity.
> We do this to avoid cluttered notations since sometimes we even need to write $Q^{\pi}(x,a; \ell)$ to specified the loss $\ell$.
> We can more clearly disambiguate the notation in Section 2 to address your concern.
>
> Q2: Adding a table of comparison.
>
> A2: This is a good idea. We will add one in the intro section if space allows.
>
> Q3: Adding experiments.
>
> A3: This will indeed help us understand the benefits of the proposed approach. We plan to do so in the full version of this work.

---

> > ### Comment · Reviewer_GhdY · 2021-08-25
> > **Discussion**
> >
> > Thank you for carefully addressing my concerns. My overall assessment of the paper remains unchanged.

---

### Official Review · Reviewer_AzMT · 2021-07-14

**Rating:** 7
**Confidence:** 4

**Summary:**

This paper makes several contributions:

1. In the episodic tabular setting, with adversarial rewards, unknown MDP dynamics and bandit feedback, it provides the first algorithm that achieves O(\sqrt T) regret bound using the regret decomposition approach, which is more computationally cheap than the approach presented in Jin et al. Typically result obtained by using this approach scales with the inverse of the visitation probability. However, here the results don't have this drawback.

2. For the linear case, the paper considers the setting with an infinite set of states and the algorithm has access to the simulator of the dynamics of MDP. Then it provides the algorithm that obtains T^2/3 regret without assuming that the smallest eigenvalue of the covariance matrix \Sigma_h is at least \lambda_min and T^1/2 taken this assumption. The latter improves the previous result in the considered setting, which was T^2/3 regret, assuming that the smallest eigenvalue of \Sigma_h is bounded away from zero. The only drawback is that the algorithm is more computationally expensive since it requires O(T^H) number of calls to the simulator.

3. The third result presented in the paper is in the same setting as above but without access to the simulator of MDP, which is a very challenging setting. The obtained regret bound is T^6/7, and it makes use of the assumption on the smallest eigenvalue of \Sigma_h.

4. All three results from above make use of dilated bonuses, which is the main contribution of this paper.
The use of the dilated bonuses that decrease the contribution to the regret, induced by the term appearing from the mismatch of q^*(x) and q(x), shows up in the analysis. To the best of my knowledge, this idea has not been considered before, and I think this idea is beneficial for future studies.


**Limitations And Societal Impact:**

There is no potential negative societal impact

**Main Review:**


- The authors state that it is important to remove the use of an exploratory policy by the algorithm. Is there an argument why it is important and why it is better to be replaced by regularization of covariance matrix? Also, is there a technical reason why implicit exploration works better?

-  Introducing the loss estimate for the tabular setting, it worth mentioning EXP3-IX algorithm for adversarial multi-armed bandits.

- Would the dilated bonuses work the same way for the REPS approach?

- Can the result of the Theorem 6.2 be improved by assuming that the comparator policy plays actions uniformly with some probability?


**Time Spent Reviewing:**

8

---

> ### Author Response · Authors · 2021-08-09
> **Response**
>
> Thanks for raising these good questions. Please see our responses below:
>
> Q1: Why is it important to remove the use of an exploratory policy?   Is there an argument why it is important and why it is better to be replaced by regularization of covariance matrix?
>
> A1: In practice, it is often not the case that a single policy can explore all directions of the feature space, let alone having access to it in advance. So many works in the reinforcement learning literature seek ways to avoid this kind of assumption. If such policy is unavailable, it means that the policy executed by the learner may not explore all directions of the feature space. This leads to difficulty in constructing unbiased loss estimators for the $Q$ functions. For example, the magnitude of $\hat{\theta}_{t,h}$ might become arbitrarily large, and adding regularization to the covariance matrix is a natural way to control this magnitude.
>
> Q2: it's worth mentioning EXP3-IX algorithm for adversarial multi-armed bandits.
>
> A2: We agree, and we will add it.
>
> Q3: Would the dilated bonuses work the same way for the REPS approach?
>
> A3: The main purpose of the dilated bonus is to add some *extra* bonus in order to let local policy search explores the global state space.
> REPS approaches perform (expensive) global optimization and obtain $\sqrt{T}$ regret already, so there is really no need to add such bonuses for these approaches.
>
> Q4: Can the result of the Theorem 6.2 be improved by assuming that the comparator policy plays actions uniformly with some probability?
>
> A4: The bound can indeed be improved if we restrict our comparator to have an uniform exploration (since in this case the corresponding exploration cost of the learner does not reflect into the regret anymore). However, we don't think this is a fair definition of regret when comparing with other works.

---

> > ### Comment · Reviewer_AzMT · 2021-09-02
> > **Response**
> >
> > Thank you for detailed answers to my questions. I keep my score unchanged.

---

### Official Review · Reviewer_9Q9q · 2021-07-16

**Rating:** 6
**Confidence:** 2

**Summary:**

The paper studies reinforcement learning in adversarial MDPs with policy optimization algorithms. Using a novel bonus design, the new algorithms achieve $\tilde{O}(S\sqrt{AT})$ regret in the tabular case, and $\tilde{O}(T^{2/3})$ regret in the linear function approximation case, assuming access to a simulator, significantly improving known results.

**Limitations And Societal Impact:**

This work is purely theoretical and do not present foreseeable negative social impact.

**Main Review:**

The key algorithmic idea of this paper is the following observation: suppose the regret with respect to $l_t-b_t$ can be bounded the loss of the optimal policy on $b_t$ (which can be large), one can cancel this term via linearity and bound the target regret with the loss of $\pi_t$ on $b_t$ (which can be controlled using standard techniques). This however requires $b_t$ to be designed in a "circular" fashion, which the authors resolve by using dilated bonuses. The new regret decomposition and bonuses are novel and interesting.

In the tabular case, the main result is an $\tilde{O}(H^2S\sqrt{AT})$ upper bound, which improves existing upper bound for policy optimization methods. Although it is one $H$ factor away from the current best result for adversarial MDP with bandit feedback [Jin et al. 2020], it enjoys a better computational efficiency. In the linear function approximation case, the first result is an $\tilde{O}(T^{2/3})$ regret bound assuming access to a simulator, which is the first known sublinear result of its kind. Although access to a simulator is usually a strong assumption, it can be justified since it is required by existing work on adversarial MDPs or even stochastic MDPs [Lattimore et al. 2020]. Other results include a $\tilde{O}(\sqrt{T})$ rate assuming an exploratory policy, and a $\tilde{O}(T^{6/7})$ rate assuming exploratory policy and linear MDP (but removing the need of a simulator). The exploratory policy is also a strong assumption, but is also assumed in existing work to achieve sublinear regret in adversarial linear MDPs. Overall the results of this paper seem to be solid contributions to the existing literature.

Other comments:

- The number of queries to the simulator is larger than $A^H$ but is not reflected in the regret bound, which is different from the stochastic settings where queries to the simulator is regarded as the sample complexity. In stochastic MDPs, with $A^H$ samples of uniform exploration, one can already evaluate any policy via Monte Carlo. The reason why this doesn't help in adversarial MDPs seems to be that the simulator does not return reward information. This makes one feel that the role of simulators in adversarial MDPs with linear function approximation (with bandits feedback) is a bit odd: the algorithm cannot extract reward information (as per Assumption 2) and cannot learn the dynamics as the state space can be infinite.

-  If the state space is finite and if one allows an infinite budget of queries, one may treat the dynamics as known and evoke existing results (e.g. [Neu et al. 2010]) whose regret bound are independent of the number of states. Thus there seems to be a trade-off between simulator query complexity and online regret, and it would be interesting to know where $A^H$ is positioned in this trade-off.

- In the algorithm box of Algorithm 2, the reference to Algorithm 6 should be Algorithm 7.

-------
Neu et al. The online loop-free stochastic shortest-path problem. 2010.

Lattimore et al. Learning with Good Feature Representations in Bandits and in RL with a Generative Model. 2020.

**Time Spent Reviewing:**

4

---

> ### Author Response · Authors · 2021-08-09
> **Response**
>
> We thank the reviewer for raising some very interesting discussions.  First, we clarify that the bound of [Neu et al., 2010] still has an implicit dependency on the number of states. Specifically, their bound has a factor $\frac{H}{\alpha}$, where $\alpha$ is such that all states can be reached by any policy with probability at least $\alpha$. By this definition, we have $\alpha\leq \min_{\pi}\min_{s}\Pr[$ Policy $\pi$ visits $s]\leq \min_{\pi}\frac{1}{|S|}\sum_{s}\Pr[$ Policy $\pi$  visits $s]= \frac{H}{|S|}$, and thus $\frac{H}{\alpha}\geq |S|$. But
> we agree that if the transition is completely known, and if the number of parameters for the losses are upper bounded by $d$, then the regret should only depend on $d$ but not $|S|$.
> (Note that our results for the linear function approximation settings have no $|S|$ dependence.)
>
> Regarding the large number of queries, we believe that it is possible to improve it to polynomial, but the answer is still highly unclear to us. In our algorithm, these queries are drawn mainly to construct the bonus function, which is just for encouraging exploration, and may not need to be constructed accurately as our algorithm does. Besides, in Section D of [Du et al., 2020], the authors described a way to get an $\epsilon$-optimal policy with $\text{poly}(d, H, 1/\epsilon)$ calls to the simulator, under only the linear-$Q$ assumption. However, their result requires that the simulator also reveals the reward information, while in our case, the learner only gets the reward information from real interactions with the environment, and can only do this once per episode. For another viewpoint, our setting is where the $Q$ function has some structure, but the transition has no (explicit) structure. In this case, whether the sampling procedure (whose goal is to learn about the transition) can benefit from the structure of the $Q$ function and reduce the number of samples is still unclear to us.
>
> [Du et al., 2020]  Simon Du, Sham Kakade, Ruosong Wang, Lin Yang. Is a good representation sufficient for sample efficient reinforcement learning? ICLR 2020.

---

### Official Review · Reviewer_9Bif · 2021-07-19

**Rating:** 7
**Confidence:** 3

**Summary:**

This paper introduces a new bonus design mechanism, and utilizes it to derive new policy optimization algorithms with improved rates for adversarial MDPs.

**Limitations And Societal Impact:**

This work is purely theoretical. I am not aware of any potential negative societal impact.


**Main Review:**

This is an interesting paper for the following reasons.

1. The paper is well written. The mechanism of the  bonus design and its technical motivation is clearly explained in Section 3.

2. Implementing the dilated bonus on tabular MDPs gives a new policy optimization algorithm with $\sqrt{T}$-regret, which improves over Shani et al., 2020. All other known algorithms with the same rate [e.g., Jin et al., 2020a] requires solving a large-scale convex optimization problem over the occupancy measure per episode, while the algorithm in this paper doesn’t because of its clever ‘local’ bonus design. I checked the proofs for Section 3&4 in details, and I believe they are correct.

3. The authors further apply their bonus technique to adversarial linear MDPs and obtain improved results over prior arts.

Overall, this paper made solid technical contributions by designing more efficient algorithms with improved rates.

Minor suggestion on related works:

The idea of adding $\gamma$ in the denominator of $\hat{Q}$ is actually first proposed in [1], which not only encourages exploration but also makes it possible to obtain high-probability bounds. If I remember correctly, [1] also proves some lemma that is highly similar to Lemma A.2 in this paper.

[1] Explore no more: Improved high-probability regret bounds for non-stochastic bandits, Gergely Neu, NIPS 2015.


**Time Spent Reviewing:**

5

---

> ### Author Response · Authors · 2021-08-09
> **Response**
>
> Thanks for the support for our work and the suggestion on adding earlier related work on the implicit exploration idea. We will do so.

---

> > ### Comment · Reviewer_9Bif · 2021-08-17
> > **Response**
> >
> > Thanks for the response. My recommendation remains acceptance.

---

### Decision · Program_Chairs · 2021-09-27

**Decision:**

Accept (Poster)

**Comment:**

The paper proposes some interesting technical tools for online learning in MDPs with adversarial rewards, improving existing results in tabular MDPs and large-scale MDPs under realizable linear function approximation and bandit feedback. All reviewers appreciated the clear improvement over previous results and the novelty of the technical contribution, with some expert reviewers opining that the proposed techniques are likely to inspire future work in the field. Based on my own reading, I concur with this assessment and agree that the paper should definitely be accepted for publication at the conference.